# Sample-Conditional Coverage in Conformal Prediction

**John Duchi**
Departments of Statistics and Electrical Engineering
Stanford University
jduchi@stanford.edu

## Abstract

We revisit the problem of constructing predictive confidence sets for which we wish to obtain some type of conditional validity. We provide new arguments showing how "split conformal" methods achieve near desired coverage levels with high probability, a guarantee conditional on the validation data rather than marginal over it. In addition, we directly consider (approximate) conditional coverage, where, e.g., conditional on a covariate $X$ belonging to some group of interest, we seek a guarantee that a predictive set covers the true outcome $Y$. We show that the natural method of performing quantile regression on a held-out (validation) dataset yields minimax optimal guarantees of coverage in these cases. Complementing these positive results, we also provide experimental evidence highlighting work that remains to develop computationally efficient valid predictive inference methods.

## 1   Introduction and background

In conformal prediction [31, 21, 22, 4], we wish to perform predictive inference on the outcome $Y$ coming from pairs $(X, Y) \in \mathcal{X} \times \mathcal{Y}$. The basic approach yields confidence sets $C(x) \subset \mathcal{Y}$, where given a sample $(X_i, Y_i)_{i=1}^n$, an estimated confidence set $\widehat{C}$ provides the (marginal) coverage

$$\mathbb{P}\left(Y_{n+1} \in \widehat{C}(X_{n+1})\right) \geq 1 - \alpha. \tag{1}$$

Typically, to do this, we assume the existence of a scoring function $s : \mathcal{X} \times \mathcal{Y} \to \mathbb{R}$ and define confidence sets of the form $C_\tau(x) \coloneqq \{y \mid s(x, y) \leq \tau\}$. For example, when predicting $Y \in \mathbb{R}$ in regression, given a predictor $f : \mathcal{X} \to \mathbb{R}$ the absolute error $s(x, y) = |f(x) - y|$ yields the familiar confidence set $C_\tau(x) = \{y \in \mathbb{R} \mid |y - f(x)| \leq \tau\} = [f(x) - \tau, f(x) + \tau]$ of values $y$ near $f(x)$.

The classical (split-conformal) approach [31, 4] uses the sample to find the threshold $\widehat{\tau}$ larger than the observed scores on about $1 - \alpha$ fraction of the data, then notes that $s(X_{n+1}, Y_{n+1})$ is likely to be smaller than this threshold. More formally, if $(X_i, Y_i)$ are exchangeable and we let $S_i = s(X_i, Y_i)$, then for the order statistics $S_{(1)} \leq S_{(2)} \leq \cdots \leq S_{(n+1)}$, we have

$$\mathbb{P}\left(S_{n+1} > S_{(\lceil (1-\alpha)(n+1) \rceil)}\right) \leq \alpha,$$

because the probability that $S_{n+1}$ is in the $\alpha$-largest fraction of the observed scores is at most $\alpha$. Then a bit of bookkeeping [e.g. 27, Lemma 2] shows that the slightly enlarged empirical quantile

$$\widehat{\tau} \coloneqq \mathsf{Quant}_{(1-\alpha)(1+1/n)}(S_1, \ldots, S_n),$$

provides the guarantee

$$\mathbb{P}\left(S_{n+1} > \widehat{\tau}\right) \leq \alpha.$$

Written differently, the confidence set

$$\widehat{C}(x) \coloneqq \{y \in \mathcal{Y} \mid s(x, y) \leq \widehat{\tau}\}$$

satisfies

$$\mathbb{P}(Y_{n+1} \in \widehat{C}(X_{n+1})) = \mathbb{P}\left(s(X_{n+1}, Y_{n+1}) \leq \widehat{\tau}\right) = \mathbb{P}\left(S_{n+1} \leq \widehat{\tau}\right) \geq 1 - \alpha.$$

## 1.1 On $X$-conditional coverage

Instead of the marginal guarantee (1), we could target conditional coverage, where we say a set valued mapping $\widehat{C}_n : \mathcal{X} \rightrightarrows \mathcal{Y}$ achieves distribution-free conditional $(1 - \alpha)$ coverage if for any $P$, when $(X_i, Y_i) \stackrel{\text{iid}}{\sim} P$ and $\widehat{C}_n$ is a function of $(X_i, Y_i)_{i=1}^n$, then for $P$-almost-all $x$,

$$\mathbb{P}(Y_{n+1} \in \widehat{C}_n(X_{n+1}) \mid X_{n+1} = x) \geq 1 - \alpha. \tag{2}$$

Vovk [30] shows this is impossible. For example, when $\mathcal{Y} = \mathbb{R}$, the Lebesgue measure $\mathsf{Leb}(\widehat{C}(x))$ is almost always infinite [30, Proposition 4] (see also extensions in [4] and [10, Corollary 7.1]):

**Corollary 1.1** ([30, 4, 10]). *Let $\mathcal{X}$ be a metric space and assume $X \in \mathcal{X}$ has continuous distribution. If $\widehat{C}$ provides distribution free $(1 - \alpha)$ conditional coverage, then for $P$-almost all $x \in \mathcal{X}$,*

$$\mathbb{P}(\mathsf{Leb}(\widehat{C}(x)) = +\infty) \geq 1 - \alpha.$$

These failures motivate relaxing the conditional coverage condition (2). The simplest approach considers *group-conditional coverage*, where for groups $G \subset \mathcal{X}$, one targets the guarantee

$$\mathbb{P}(Y_{n+1} \in \widehat{C}(X_{n+1}) \mid X_{n+1} \in G) \geq 1 - \alpha. \tag{3}$$

Barber et al. [4, Sec. 4] achieve the coverage (3) by considering worst-case coverage over groups $G$; Jung et al. [17] provide variations. Gibbs, Cherian, and Candès [12] extend this idea, beginning by observing that conditional coverage $\mathbb{P}(Y \in \widehat{C}(x) \mid X = x) = 1 - \alpha$ holds if and only if

$$\mathbb{E}\left[w(X)\left(1\left\{Y \in \widehat{C}(X)\right\} - (1 - \alpha)\right)\right] = 0 \tag{4}$$

for all bounded $w$. Similarly, the one-sided inequality (2) holds if and only if

$$\mathbb{E}\left[w(X)1\left\{Y \in \widehat{C}(X)\right\}\right] \geq (1 - \alpha)\mathbb{E}[w(X)]$$

for all nonnegative bounded $w$. Taking $w(x) = 1\{x \in G\}$ for groups $G \subset \mathcal{X}$ implies the group-conditional coverage (3); relaxing the condition (4) by considering subclasses of weighting functions $\mathcal{W} \subset \{\mathcal{X} \to \mathbb{R}\}$ leads to the following definition [12]:

**Definition 1.1** (Gibbs et al. [12]). *A confidence set $C : \mathcal{X} \rightrightarrows \mathcal{Y}$ achieves $\mathcal{W}$-weighted $((1 - \alpha), \epsilon)$ coverage if*

$$\left|\mathbb{E}\left[w(X)\left(1\{Y \in C(X)\} - (1 - \alpha)\right)\right]\right| \leq \epsilon \text{ for all } w \in \mathcal{W}.$$

Gibbs et al.'s main two examples take $\mathcal{W}$ of the form $\mathcal{W} = \{w \mid w(x) = \langle v, \phi(x) \rangle\}$ for some feature mapping $\phi : \mathcal{X} \to \mathbb{R}^d$ or to correspond to a reproducing kernel Hilbert space. On a new example $X_{n+1}$ they perform *full conformal inference* [31], where implicitly for each $t \in \mathbb{R}$, they solve

$$\widehat{h}_{n+1,t} = \underset{h \in \mathcal{W}}{\arg\min} \sum_{i=1}^n \ell_\alpha(h(X_i) - S_i) + \ell_\alpha(h(X_{n+1}) - t)$$

for the quantile loss $\ell_\alpha(t) = \alpha \left[t\right]_+ + (1 - \alpha) \left[-t\right]_+$, then define the implicit confidence set

$$\widehat{C}_n(X_{n+1}) := \left\{y \in \mathcal{Y} \mid s(X_{n+1}, y) \leq \widehat{h}_{n+1, s(X_{n+1}, y)}(X_{n+1})\right\}. \tag{5}$$

A careful duality calculation [12, Sec. 4] shows how to compute $\widehat{C}_n$ by solving a linear program over $O(n + d)$ variables using $(X_i)_{i=1}^{n+1}$ and $S_1^n = (S_1, \ldots, S_n)$, and Gibbs et al. show the set (5) satisfies

$$\left|\mathbb{E}\left[w(X_{n+1})1\left\{Y_{n+1} \notin \widehat{C}_n(X_{n+1})\right\} - w(X_{n+1})(1 - \alpha)\right]\right| \leq \epsilon_{\text{int}}(w)$$

for $w \in \mathcal{W}$, where $\epsilon_{\text{int}}(w)$ is a small interpolation error term. Defining $\mathbb{P}_w(A) = \mathbb{E}_P[w(X)1\{A\}]/\mathbb{E}_P[w(X)]$ to be the $w$-weighted probability of an event $A$ for $w \geq 0$, this inequality strengthens inequality (1) to imply that for all $w \geq 0, w \in \mathcal{W}$,

$$\mathbb{P}_w(Y_{n+1} \in \widehat{C}(X_{n+1})) \geq 1 - \alpha - \epsilon_{\text{int}}(w). \tag{6}$$

Computing this prediction set $\widehat{C}_n$ requires solving a sometimes costly optimization. This suggests split-conformal approaches that provide adaptive confidence sets of the form

$$\widehat{C}_n(x) := \left\{ y \in \mathcal{Y} \mid s(x,y) \le \widehat{h}_n(x) \right\},$$

where $\widehat{h}_n$ is chosen based only on the sample $(X_i, Y_i)_{i=1}^n$, making the set $\widehat{C}_n$ easy to compute [27, 8]. In spite of their ease of computation, it has been challenging to demonstrate that these sets can achieve coverage; for example, Romano et al. [27] and Cauchois et al. [8] apply another level of conformalization to fit a constant threshold $\widehat{\tau}_n$ and use $\widehat{C}_n(x) = \{ y \in \mathcal{Y} \mid s(x,y) \le \widehat{h}_n(x) + \widehat{\tau}_n \}$. We show new coverage guarantees for these sets, including new optimality guarantees.

## 1.2  Sample-conditional coverage

Inequalities (1) and (6) provide guarantees marginal over the entire procedure drawing $(X_i, Y_i) \overset{\text{iid}}{\sim} P$, $i = 1, \ldots, n+1$. While conditional coverage (on $X_{n+1}$) is impossible, it is possible to achieve *sample-conditional*-coverage. Letting $P_n$ denote the empirical distribution of $(X_i, Y_i)_{i=1}^n$ and recalling the weighted probability (6), our results demonstrate that with high probability over the sample $P_n$,

$$\mathbb{P}_w(Y_{n+1} \in \widehat{C}_n(X_{n+1}) \mid P_n) \ge 1 - \alpha - O(1) \sqrt{\frac{\alpha(1-\alpha)}{\mathbb{E}_P[w(X)]} \cdot \frac{d \log n}{n}}$$

simultaneously for all $w \ge 0$ in $d$-dimensional classes of functions $\mathcal{W}$. Because of their reliance on individual examples, full-conformal procedures cannot achieve such conditional guarantees [6].

Sample-conditional results do hold for split-conformal procedures in the special case that $\mathcal{W}$ consists of the constant function $w = 1$, which we now review. To state things formally, let $S_i = s(X_i, Y_i)$, where $(X_i, Y_i) \overset{\text{iid}}{\sim} P$. Let $\alpha \in (0,1)$ be a desired confidence level, and define the empirical $(1 - \alpha)$ quantile

$$\widehat{\tau}_n := \inf \left\{ t \in \mathbb{R} \mid P_n(S \le t) \ge 1 - \alpha \right\},$$

where $P_n$ denotes the empirical distribution. Given this quantile, define the confidence set

$$\widehat{C}_n(x) := \{ y \in \mathcal{Y} \mid s(x,y) \le \widehat{\tau}_n \}.$$

**Proposition 1** (Vovk [30], Proposition 2). *Let the construction above hold. Then for any $\gamma > 0$, with probability at least $1 - e^{-2n\gamma^2}$ over the sample $P_n$,*

$$\mathbb{P}(Y_{n+1} \in \widehat{C}_n(X_{n+1}) \mid P_n) \ge 1 - \alpha - \gamma. \tag{7}$$

Jung et al. [17] consider attaining sample- and group-conditional coverage (3), showing that it is approximately achievable when one assumes quantitative smoothness of the distribution of $s(x, Y)$ given $X = x$. Section 2 revisits their approach, providing sharper guarantees without any assumptions on the underlying distribution, yielding approximate conditional analogues of the guarantee (7).

## 1.3  Sample conditional coverage revisited

We begin by revisiting the sample-conditional coverage guarantees of Proposition 1. In the appendices (see Appendix A), we provide an elementary proof relying only on Hoeffding's concentration inequality, along with two other proofs reposing on uniform convergence to form the point of departure for our more sophisticated coverage guarantees. In the interest of brevity, we focus on new results, beginning with a Bernstein-type guarantee that more carefully tracks the desired coverage $\alpha$:

**Proposition 2.** *Let $\delta \in (0, 1)$ and define*

$$\gamma_n(\delta) := \frac{4 \log \frac{1}{\delta}}{3n} + \sqrt{\left( \frac{4}{3n} \log \frac{1}{\delta} \right)^2 + \frac{2\alpha(1-\alpha)}{n} \log \frac{1}{\delta}} \le \frac{8 \log \frac{1}{\delta}}{3n} + \sqrt{\frac{2\alpha(1-\alpha)}{n} \log \frac{1}{\delta}}.$$

*Then with probability at least $1 - \delta$ over the draw of the sample $P_n$,*

$$1 - \alpha - \gamma_n(\delta) \le \mathbb{P}(Y_{n+1} \in \widehat{C}_n(X_{n+1}) \mid P_n).$$

*If additionally the scores $S$ have a density, then with probability at least $1 - 2\delta$,*

$$1 - \alpha - \gamma_n(\delta) \le \mathbb{P}(Y_{n+1} \in \widehat{C}_n(X_{n+1}) \mid P_n) \le 1 - \alpha + \gamma_n(\delta).$$

We see that the quantile-based confidence set achieves coverage $1 - \alpha \pm O(1)\sqrt{\alpha(1-\alpha)/n}$. When $\alpha$ is small—which is the typical case—this is always sharper than the naive guarantee (7). The central limit theorem shows this is as accurately as we could hope to even estimate the coverage level of a predictor; moreover, as we discuss following Theorem 3, it is minimax (rate) optimal. In Appendix A, we provide two proofs of Proposition 1, using Appendix A.3 to prove Proposition 2.

## 2   Approaching conditional coverage

Keeping in mind the ideas in Sections 1.1 and 1.2, we revisit conditional coverage, but we do so conditional on the sample $P_n$ as well. To do this, we consider Jung et al. and Gibbs et al.'s approaches [17, 12], considering quantile estimation via the quantile loss [18], which for $\alpha > 0$ is

$$\ell_\alpha(t) := \alpha\,[t]_+ + (1-\alpha)\,[-t]_+\,.$$

For a random variable $Y$, the quantile $\mathsf{Quant}_{1-\alpha}(Y) := \inf\{t \mid \mathbb{P}(Y \le t) \ge 1 - \alpha\}$ minimizes $L(t) := \mathbb{E}[\ell_\alpha(t - Y)]$. Now (cf. [12, 17]), consider a quantile regression of attempting to predict $S \in \mathbb{R}$ from $X \in \mathcal{X}$. Let $\phi : \mathcal{X} \to \mathbb{R}^d$ be a feature mapping and consider $L(\theta) := \mathbb{E}[\ell_\alpha(\langle \theta, \phi(X)\rangle - S)]$. Then (for motivation) assuming that $S$ has a density conditional on $X = x$, we see that

$$\nabla L(\theta) = \mathbb{E}\left[(1\{S < \langle \theta, \phi(X)\rangle\} - (1-\alpha))\,\phi(X)\right].$$

Now let $\theta^\star \in \operatorname{argmin} L(\theta)$ be a population minimizer. Then, as Gibbs et al. [12] note, for *any* $u \in \mathbb{R}^d$, we have

$$0 = \langle u, \nabla L(\theta^\star)\rangle = \mathbb{E}\left[(\mathbb{P}(S < \langle \theta^\star, \phi(X)\rangle \mid X) - (1-\alpha)) \cdot \langle u, \phi(X)\rangle\right].$$

This connects transparently to confidence set mappings [12, 17]: taking

$$S = s(X, Y) \ \text{ and } \ C_{\theta^\star}(x) := \{y \in \mathcal{Y} \mid s(x, y) \le \langle \theta^\star, \phi(X)\rangle\}\,,$$

we have

$$0 = \mathbb{E}\left[(\mathbb{P}(Y \in C(X) \mid X) - (1-\alpha))\,\langle u, \phi(X)\rangle\right] \ \text{ for all } u \in \mathbb{R}^d.$$

In turn, Gibbs et al. [12] show this implies the population coverage guarantee

**Corollary 2.1** (Gibbs et al. [12], Thm. 2). *Assume the distribution of $S \mid X$ is continuous. Let $\theta^\star$ minimize $L(\theta) = \mathbb{E}[\ell_\alpha(\langle \theta, \phi(X)\rangle - s(X, Y))]$. Then $C_{\theta^\star}$ provides $((1 - \alpha), 0)$-weighted coverage (Definition 1.1) for the class $\mathcal{W} := \{w(x) = \langle u, \phi(x)\rangle\}_{u \in \mathbb{R}^d}$ of linear functions of $\phi(x)$.*

We extend these guarantees to provide sample-conditional coverage by adapting the arguments we use to prove Proposition 1.

### 2.1   An estimated confidence set

The population-level confidence set $C_{\theta^\star}(x) = \{y \mid s(x, y) \le \langle \theta^\star, \phi(x)\rangle\}$ immediately suggests the empirical analogue

$$\widehat{\theta}_n \in \operatorname*{argmin}_\theta \mathbb{E}_{P_n}\left[\ell_\alpha(\langle \theta, \phi(X)\rangle - S)\right], \tag{8}$$

which Jung et al. [17] consider for the special case that the feature mapping $\phi(x) = [1\{x \in G\}]_{G \in \mathcal{G}}$ is an indicator vector for groups $G \subset \mathcal{X}$. This gives the confidence set

$$\widehat{C}_n(x) := \left\{y \in \mathcal{Y} \mid s(x, y) \le \langle \widehat{\theta}, \phi(x)\rangle\right\}.$$

This set indeed provides sample-conditional coverage. To see this, assume for simplicity that $\phi(x)$ satisfies $\|\phi(x)\|_2 \le b_\phi$ for all $x$, and let $\mathbb{B}_2 = \{u \in \mathbb{R}^d \mid \|u\|_2 \le 1\}$ be the $\ell_2$-ball.

**Theorem 1.** *Assume the boundedness conditions above and that $n \ge d$. Let $\widehat{\theta}$ be the empirical minimizer (8) of the $\alpha$-quantile loss, let $\widehat{h}(x) = \langle \widehat{\theta}, \phi(x)\rangle$, and define the confidence set*

$$\widehat{C}_n(x) := \left\{y \in \mathcal{Y} \mid s(x, y) \le \widehat{h}(x)\right\}.$$

*Then there exists a constant $c \leq 2 + \alpha/\sqrt{d}$ such that for $t \geq 0$, with probability at least $1 - e^{-nt^2}$ over the draw of the sample $P_n$,*

$$\mathbb{E}\left[\langle u, \phi(X_{n+1})\rangle \left(1\left\{Y_{n+1} \in \widehat{C}_n(X_{n+1})\right\} - (1-\alpha)\right) \mid P_n\right] \geq -cb_\phi \left(\sqrt{\frac{d}{n}\log\frac{n}{d}} + t\right)$$

*simultaneously for all $u \in \mathbb{B}_2$ satisfying $\langle u, \phi(x)\rangle \geq 0$ for all $x \in \mathcal{X}$. If the scores $S_i$ are distinct with probability 1, then with the same probability,*

$$\mathbb{E}\left[\langle u, \phi(X_{n+1})\rangle \left(1\left\{Y_{n+1} \in \widehat{C}_n(X_{n+1})\right\} - (1-\alpha)\right) \mid P_n\right] \leq 3b_\phi \left(\sqrt{\frac{d}{n}\log\frac{n}{d}} + t + \frac{d}{3n}\right).$$

*simultaneously for all $u \in \mathbb{B}_2$.*

We defer the proof of Theorem 1 to Section 2.2. We also note in passing that by randomizing, it is possible to make the scores distinct without sacrificing coverage [cf. 12].

As a corollary to Theorem 1, assume that $\mathcal{G} = \{G_1, \ldots, G_d\}$ partitions $\mathcal{X}$, and define the feature indicator $\phi_{\mathcal{G}}(x) = (1, 1\{x \in G_1\}, \ldots, 1\{x \in G_d\})$. With this choice, we obtain

**Corollary 2.2.** *Let $\widehat{\theta}$ be as in Theorem 1 and $\phi = \phi_{\mathcal{G}}$ be the group feature function. Then simultaneously for all groups $G_j$,*

$$\mathbb{P}(Y_{n+1} \in \widehat{C}_n(X_{n+1}) \mid X_{n+1} \in G_j, P_n) \geq 1 - \alpha - \frac{4}{\mathbb{P}(X_{n+1} \in G_j)}\left(\sqrt{\frac{d}{n}\log\frac{n}{d}} + t\right)$$

*with probability at least $1 - e^{-nt^2}$.*

We will sharpen this inequality via more sophisticated arguments in the sequel.

It is instructive, however, to compare this guarantee to that the full-conformal approach (5) provides. The construction (5) appears to obtain better coverage than the more basic approaches here [12, Fig. 3], but it can be more computationally challenging [12, Fig. 6]. Gibbs et al. [12] show that a randomized version of their procedure (5) achieves

$$\mathbb{P}(Y_{n+1} \in \widehat{C}_n(X_{n+1}) \mid X_{n+1} \in G) = 1 - \alpha \text{ for all } G \in \mathcal{G}.$$

This can be substantially sharper than the guarantee Corollary 2.2 provides, as our sample-conditional coverage guarantees are not quite so exact; we revisit these points in experiments.

## 2.2 Proof of Theorem 1

By convexity, $0 \in \sum_{i=1}^n \partial\ell_\alpha(\langle\widehat{\theta}, \phi(X_i)\rangle - S_i)\phi(X_i)$, which is equivalent to the statement that for some scalars (really, dual variables) $g_i$ satisfying

$$g_i \begin{cases} = \alpha & \text{if } \langle\widehat{\theta}, \phi(X_i)\rangle > S_i \\ = -(1-\alpha) & \text{if } \langle\widehat{\theta}, \phi(X_i)\rangle < S_i \\ \in [-(1-\alpha), \alpha] & \text{if } \langle\widehat{\theta}, \phi(X_i)\rangle = S_i \end{cases} \tag{9}$$

we have $0 = \sum_{i=1}^n g_i\phi(X_i)$. We use the empirical process notation $P_n f = \frac{1}{n}\sum_{i=1}^n f(X_i)$ for shorthand. Recall that for a convex function $f$, the directional derivative $f'(x; u) = \lim_{t\downarrow 0}\frac{f(x+tu)-f(x)}{t}$ exists and satisfies $f'(x; u) = \sup\{\langle g, u\rangle \mid g \in \partial f(x)\}$. Thus, by definition of optimality,

$$P_n\ell'_\alpha(\langle\widehat{\theta}, \phi(X)\rangle - S; u) \geq 0$$

for all $u$. Let $u$ be such that $\langle u, \phi(x)\rangle \geq 0$ for all $x \in \mathcal{X}$. Then

$$\ell'_\alpha(\langle\theta, \phi(x)\rangle - s; u) = \langle\phi(x), u\rangle\left[\alpha 1\{\langle\theta, \phi(x)\rangle \geq s\} - (1-\alpha)1\{\langle\theta, \phi(x)\rangle < s\}\right].$$

By the first-order optimality condition we obtain

$$0 \leq \left\langle u, \alpha P_n\phi(X)1\left\{\langle\widehat{\theta}, \phi(X)\rangle \geq S\right\} - (1-\alpha)P_n\phi(X)1\left\{\langle\widehat{\theta}, \phi(X)\rangle < S\right\}\right\rangle$$

$$= \left\langle u, \alpha P_n\phi(X) - P_n\phi(X)1\left\{\langle\widehat{\theta}, \phi(X)\rangle < S\right\}\right\rangle.$$

Suppose that we demonstrate that

$$\left\| \frac{1}{n} \sum_{i=1}^{n} \phi(X_i) 1\{S_i > \langle \theta, \phi(X_i) \rangle\} - \mathbb{E}_P[\phi(X) 1\{S > \langle \theta, \phi(X) \rangle\}] \right\|_2 \le \epsilon$$

uniformly over $\theta \in \mathbb{R}^d$. Then we would obtain for all $u \in \mathbb{B}_2$ with $\langle u, \phi(x) \rangle \ge 0$ for all $x \in \mathcal{X}$,

$$0 \le \left\langle u, P_n \phi(X) \left( \alpha - 1\{S > \langle \widehat{\theta}, \phi(X) \rangle\} \right) \right\rangle \le \mathbb{E}_P \left[ \langle u, \phi(X) \rangle \left( \alpha - 1\{S > \langle \widehat{\theta}, \phi(X) \rangle\} \right) \right] + \epsilon.$$

That is, as $y \notin \widehat{C}(x)$ if and only if $s(x, y) > \langle \widehat{\theta}, \phi(x) \rangle$,

$$\mathbb{E}_P \left[ \langle u, \phi(X) \rangle \left( 1\{Y \notin \widehat{C}(X)\} - \alpha \right) \right] \le \epsilon. \tag{10}$$

With appropriate $\epsilon$, this will give the first claim of the theorem.

We abstract a bit and let $\mathcal{H} \subset \{\mathcal{X} \to \mathbb{R}\}$ be a collection of functions, and consider the process

$$Z_n(h) := \frac{1}{n} \sum_{i=1}^{n} \phi(X_i) 1\{S_i > h(X_i)\}.$$

When $\mathcal{H}$ is a VC-class, for each coordinate $j$, functions of the form $\phi_j(x) 1\{s > h(x)\}$ are VC-subgraph [29, Lemma 2.6.18]. The following technical lemma, whose proof we provide in Appendix C.1, shows that $Z_n$ concentrates.

**Lemma 2.1.** *Let $\mathbb{B}_2 = \{u : \|u\|_2 \le 1\}$ and $\mathcal{H}$ have VC-dimension $k$. Then*

$$\mathbb{E} \left[ \sup_{h \in \mathcal{H}, u \in \mathbb{B}_2} \langle u, Z_n(h) - \mathbb{E}[Z_n(h)] \rangle \right] \le 2 \sqrt{\frac{k \log \frac{ne}{k}}{n}} \mathbb{E} \left[ \frac{1}{n} \sum_{i=1}^{n} \|\phi(X_i)\|^2 \right]^{1/2}.$$

Trivially, $\mathbb{E}[\sup_{u \in \mathbb{B}_2} \langle u, P_n \phi(X) - \mathbb{E}[\phi(X)] \rangle] \le \frac{1}{\sqrt{n}} \mathbb{E}[\|\phi(X)\|_2^2]^{1/2}$ addressing $\alpha P_n \phi(X)$ terms.

We can extend the lemma by homogeneity to capture arbitrary vectors. Note that if we change a single example $(X_i, S_i)$, then $\langle u, Z_n(h) \rangle$ changes by at most $n^{-1} \sup_x \langle u, \phi(x) \rangle \le n^{-1} \|u\|_2 \sup_x \|\phi(x)\|_2$. By homogeneity, for any scalar $t$ there exists $u \in \mathbb{R}^d$ such that $\langle u, Z_n(h) - \mathbb{E}[Z_n(h)] \rangle \ge \|u\|_2 t$ if and only if there exists $u \in \mathbb{S}^{d-1}$ such that $\langle u, Z_n(h) - \mathbb{E}[Z_n(h)] \rangle \ge t$. So if $b_\phi = \sup_{x \in \mathcal{X}} \|\phi(x)\|_2$, we obtain by bounded differences (Lemma A.1) and homogeneity that

$$\mathbb{P} \left( \sup_{u \ne 0, h \in \mathcal{H}} \frac{\langle u, Z_n(h) - \mathbb{E}[Z_n(h)] \rangle}{\|u\|_2} \ge b_\phi t + \mathbb{E} \left[ \sup_{u \in \mathbb{S}^{d-1}, h \in \mathcal{H}} \langle u, Z_n(h) - \mathbb{E}[Z_n(h)] \rangle \right] \right) \le e^{-nt^2}.$$

Summarizing, we have proved the following proposition.

**Proposition 3.** *Let $\mathcal{H}$ have VC-dimension $k$. Then for $t \ge 0$,*

$$\mathbb{P} \left( \sup_{u \ne 0, h \in \mathcal{H}} \frac{\langle u, Z_n(h) - \mathbb{E}[Z_n(h)] \rangle}{\|u\|_2} \ge 2 b_\phi \sqrt{\frac{k \log \frac{n}{k}}{n}} + b_\phi t \right) \le e^{-nt^2}.$$

By taking $\mathcal{H} = \{h : h(x) = \langle \theta, \phi(x) \rangle\}_{\theta \in \mathbb{R}^d}$, which has VC-dimension $d$, in Proposition 3, we have thus shown that inequality (10) holds with

$$\epsilon = 2 b_\phi \sqrt{\frac{d \log \frac{n}{d}}{n}} + b_\phi t + \frac{b_\phi \alpha}{\sqrt{n}}$$

with probability at least $1 - e^{-nt^2}$, which is the first claim of Theorem 1.

We turn to the second claim of Theorem 1, which applies when $S_i$ are distinct with probability 1. Recall the definition (9) of the subgradient terms $g_i$ and define the sets $\mathcal{I}_+ = \{i \mid \langle \widehat{\theta}, \phi(X_i) \rangle > S_i\}$, $\mathcal{I}_- = \{i \mid \langle \widehat{\theta}, \phi(X_i) \rangle < S_i\}$, and $\mathcal{I}_0 = \{i \mid \langle \widehat{\theta}, \phi(X_i) \rangle = S_i\}$. Then

$$\sum_{i=1}^{n} \phi(X_i) \left( 1\{S_i > \langle \widehat{\theta}, \phi(X_i) \rangle\} - \alpha \right)$$

$$= - \sum_{i \in \mathcal{I}_+ \cup \mathcal{I}_-} \phi(X_i) g_i - \sum_{i \in \mathcal{I}_0} \phi(X_i) g_i - \sum_{i \in \mathcal{I}_0} \phi(X_i) (\alpha - g_i) = \sum_{i \in \mathcal{I}_0} (g_i - \alpha) \phi(X_i),$$

where we used that $\sum_{i=1}^n g_i \phi(X_i) = 0$ by construction. Now, we leverage our assumption that $S_i$ are distinct with probability 1. We see immediately that $\operatorname{card}(\mathcal{I}_0) \leq d$, because with distinct values $S_i$ we may satisfy (at most) $d$ linear equalities, and so

$$\left\| \frac{1}{n} \sum_{i=1}^n \phi(X_i) \left( 1\{ S_i > \langle \widehat{\theta}, \phi(X_i) \rangle \} - \alpha \right) \right\|_2 \leq \frac{\operatorname{card}(\mathcal{I}_0)}{n} b_\phi \leq \frac{d}{n} b_\phi.$$

Because Proposition 3 controls the fluctuations of the process $\theta \mapsto \phi(x) 1\{s > \langle \phi(x), \theta \rangle\}$, we obtain that with probability at least $1 - e^{-nt^2}$,

$$\left\| \mathbb{E}\left[ \phi(X) 1\{ S > \langle \widehat{\theta}, \phi(X) \rangle \} \right] - \alpha \mathbb{E}[\phi(X)] \right\|_2$$
$$\leq \left\| \frac{1}{n} \sum_{i=1}^n \phi(X_i) \left( 1\{ S_i > \langle \widehat{\theta}, \phi(X_i) \rangle \} - \alpha \right) \right\|_2 + 2 b_\phi \sqrt{\frac{d}{n} \log \frac{n}{d}} + \frac{b_\phi \alpha}{\sqrt{n}} + b_\phi t$$
$$\leq b_\phi \frac{d}{n} + 3 b_\phi \sqrt{\frac{d}{n} \log \frac{n}{d}} + b_\phi t.$$

## 3  Sharper and rate-optimal approximate conditional bounds

The bounds Theorem 1 provides do not reflect the sharpest coverage possible. By leveraging empirical process variants of the Bernstein concentration inequalities we use to prove Proposition 2, we can achieve sharper bounds on weighted coverage that adapt to the linear functionals $x \mapsto w(x) = \langle u, \phi(x) \rangle$. As a consequence of our results, in terms of achieving approximate conditional coverage (i.e., weighted coverage as in Definition 1.1), the empirical estimator (8) is minimax rate optimal; we discuss this after Theorem 3.

To state our results, assume that $\mathbb{B} \subset \mathbb{R}^d$ is an arbitrary but bounded set of vectors, and define

$$b_\phi(u) := \sup_{x \in \mathcal{X}} |\langle u, \phi(x) \rangle| \quad \text{and} \quad b_\phi := \sup_{u \in \mathbb{B}} b_\phi(u).$$

We can then extend Proposition 2 to weighted conditional coverage (Def. 1.1), conditional on the sample. We defer their proofs, presenting the building blocks common to both in Appendix B.1, then specializing in Appendicess B.2 and B.3, respectively.

**Theorem 2.** *Let $K_n = 1 + \log_2 n$. Then there exists a numerical constant $c < \infty$ such that for all $t \geq 0$, with probability at least $1 - 2 K_n e^{-t} - e^{-d \log n - t}$,*

$$\mathbb{E}\left[ \langle u, \phi(X_{n+1}) \rangle \left( 1\{ Y_{n+1} \notin \widehat{C}(X_{n+1}) \} - \alpha \right) \mid P_n \right]$$
$$\leq c \left[ \sqrt{b_\phi(u) \alpha \cdot \mathbb{E}[\langle u, \phi(X) \rangle]} \sqrt{\frac{d \log n + t}{n}} + b_\phi \frac{d \log n + t}{n} \right]$$

*simultaneously for all $u \in \mathbb{B}$ such that $\langle u, \phi(x) \rangle \geq 0$ for all $x$. If additionally the scores $S_i$ are distinct with probability 1, then with the same probability,*

$$\mathbb{E}\left[ \langle u, \phi(X_{n+1}) \rangle \left( 1\{ Y_{n+1} \notin \widehat{C}(X_{n+1}) \} - \alpha \right) \mid P_n \right]$$
$$\geq -c \left[ \sqrt{b_\phi(u) \alpha \cdot \mathbb{E}[\langle u, \phi(X) \rangle]} \sqrt{\frac{d \log n + t}{n}} + b_\phi \frac{d \log n + t}{n} \right]$$

*simultaneously for all $u \in \mathbb{B}$ such that $\langle u, \phi(x) \rangle \geq 0$ for all $x$.*

Simplifying the statement and ignoring higher-order terms, we can obtain a guarantee for weighted coverage (6): for the class $\mathcal{W} = \{ w(x) = \langle u, \phi(x) \rangle \}_{u \in \mathbb{R}^d}$, with probability $1 - e^{-t}$,

$$\mathbb{P}_w(Y_{n+1} \in \widehat{C}_{n+1} \mid P_n) \geq 1 - \alpha - O(1) \left[ \sqrt{\frac{\alpha}{\mathbb{E}[w(X)]}} \cdot \sqrt{\frac{d \log n + t}{n}} \right]$$

simultaneously for $w \geq 0$ with the normalization that $w(x) = \langle u, \phi(x) \rangle$ for a $u$ satisfying $b_\phi(u) = 1$.

Applying the theorem to group indicators, meaning that we have a collection of groups $\mathcal{G} \subset 2^{\mathcal{X}}$, and the feature mapping $\phi(x) = (1\{ x \in G \})_{G \in \mathcal{G}}$, we have the following sharpening of Theorem 1.

**Corollary 3.1.** *Assume that $\phi(x) = (1\{x \in G\})_{G \in \mathcal{G}}$, and let $d = \text{card}(\mathcal{G})$. Then with probability at least $1 - 3\log_2 ne^{-t}$,*

$$\mathbb{P}(Y_{n+1} \notin \widehat{C}(X_{n+1}) \mid X_{n+1} \in G, P_n) \le \alpha + c\left[\sqrt{\frac{\alpha}{\mathbb{P}(X_{n+1} \in G)}\frac{d\log n + t}{n}} + \frac{d\log n + t}{\mathbb{P}(X_{n+1} \in G) \cdot n}\right]$$

*simultaneously for all $G \in \mathcal{G}$.*

The result follows immediately upon considering the standard basis vectors $u = e_i$.

When the scores $S = s(X, Y)$ are distinct with probability 1, we achieve two sided bounds extending Theorem 2, as in Proposition 2. The next theorem provides an exemplar result.

**Theorem 3.** *Let the conditions of Theorem 2 hold, except assume that $S_i$ are distinct with probability 1, and that the mapping $\phi(x)$ includes a constant bias term $\phi_1(x) = 1$. Then there exists a numerical constant $c < \infty$ such that for all $t \ge 0$, with probability at least $1 - 2K_n e^{-t} - e^{-d\log n - t}$, simultaneously for all $u \in \mathbb{B}$,*

$$\left|\mathbb{E}\left[\langle u, \phi(X)\rangle \left(1\{Y_{n+1} \notin \widehat{C}(X_{n+1})\} - \alpha\right) \mid P_n\right]\right| \le c\left[b_\phi(u)\sqrt{\alpha}\sqrt{\frac{d\log n + t}{n}} + b_\phi\frac{d\log n + t}{n}\right].$$

The conclusion is weaker than that of Theorem 2, as it replaces $\sqrt{b_\phi(u)\mathbb{E}[\langle u, \phi(X)\rangle]}$ with $b_\phi(u)$.

## 3.1 Minimax optimality

The convergence guarantees in Theorems 2 and 3 are sharp to within logarithmic factors, and capture the correct dependence on $\alpha$ and the weight functions $\mathcal{W}$. Indeed, Areces et al. [1] develop a set of lower bounds that apply to VC-classes of functions $\mathcal{W}$, where they show the following. Assume that the estimated confidence set $\widehat{C}_n$ takes the form $\widehat{C}_n(x) = \{y \mid s(x, y) \le \widehat{h}(x)\}$ or $\widehat{C}_n(x) = \{y \mid \widehat{a}(x) \le s(x, y) \le \widehat{b}(x)\}$ for *some* estimated functions $\widehat{h}, \widehat{a}$, or $\widehat{b}$.

**Corollary 3.2** (Areces et al. [1], Thm. 1). *Let $\mathcal{W}$ be any class of functions mapping $\mathcal{X}$ to $\{\pm 1\}$ with VC-dimension $d$, $\widehat{C}_n$ take either of the forms above. Then there exists a sampling distribution $P$ for which $S \mid X$ has continuous bounded density and such that with constant probability over the draw of $P_n$,*

$$\left|\mathbb{E}\left[w(X_{n+1})1\{Y_{n+1} \notin \widehat{C}_n(X_{n+1})\} - w(X_{n+1})\alpha \mid P_n\right]\right| \ge c\sqrt{\frac{d\alpha(1 - \alpha)}{n}},$$

*where $c > 0$ is a universal constant.*

To compare this with Theorems 2 and 3, let $\{G_1, \ldots, G_d\}$, $G_j \subset \mathcal{X}$, be a partition of $\mathcal{X}$ into $d$ groups, and define the group feature mapping $\phi(x) = [1\{x \in G_j\}]_{j=1}^d$. Then the class of linear functionals $\mathcal{W} = \{w \mid w(x) = \langle u, \phi(x)\rangle\}_{u \in \mathbb{R}^d}$ has VC-dimension $d$, as does its restriction $\mathcal{W}_1 = \{w \mid w(x) = \langle u, \phi(x)\rangle\}_{u \in \{\pm 1\}^d}$, where $w \in \mathcal{W}_1$ satisfies $w(x) \in \{\pm 1\}$. Theorems 2 and 3, conversely, demonstrate that for $\mathbb{B}_1 = \{u \mid \|u\|_1 \le 1\}$, we have

$$\left|\mathbb{E}\left[w(X_{n+1})\left(1\{Y_{n+1} \notin \widehat{C}_n(X_{n+1})\} - \alpha\right) \mid P_n\right]\right| \le c\sqrt{\frac{\alpha(1 - \alpha)}{n}} \cdot \sqrt{d\log n + t}$$

with probability at least $1 - e^{-t}$ simultaneously for all $w$ of the form $w(x) = \langle u, \phi(x)\rangle$ for some $u \in \mathbb{B}_1$, as long as $\alpha < \frac{1}{2}$ (where we use $1 - \alpha \ge \frac{1}{2}$ and wrap constants together for a cleaner statement). In contrast to existing results [17, 1], these new guarantees are evidently optimal.

## 4 Experimental Results

Our main purpose has been to investigate conditional quantile estimation procedures, providing theoretical bounds for their performance; there is already practical experience with these methods [12]. We thus provide an exploratory experiment on the CIFAR-100 dataset [20], a 100-class image classification dataset consisting of 60,000 training examples and a 10,000 example test set, highlighting

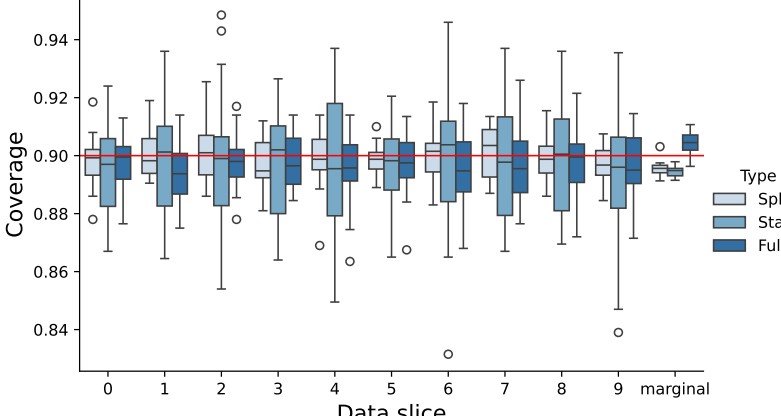

**Figure 1.** Coverage of full conformal, split conformal, and static split conformal methods on random 20% "slices" of CIFAR-100 data.

that these conditional approaches can provide better coverage than using a static threshold, i.e., $\widehat{C}_n(x) = \{y \in \mathcal{Y} \mid s(x, y) \leq \widehat{\tau}_n\}$. In Appendix D we provide a few further simulations to investigate heuristic corrections to the nominal level $\alpha$ that may yield better realized coverage.

We use the output features of a 50 layer ResNet, pre-trained on ImageNet [13, 14], as $d = 2048$-dimensional input to a 100 class logistic regression. We repeat the following experiment 10 times:

1. Uniformly randomly split the training examples into a validation set of size 10,000 and a model training set of size 50,000, on which we fit a linear classifier $s : \mathbb{R}^d \to \mathbb{R}^k$, where $s_y(x) = \langle \beta_y, x \rangle$ is the score assigned to class $y$, using multinomial logistic regression.

2. Draw a random matrix $W \in \mathbb{R}^{d \times d_0}$, where $d_0 = 10$ and $W_{ij} \overset{\text{iid}}{\sim} \mathsf{N}(0, 1)$, and use the validation data with score function $s(x, y) = s_y(x)$ and the lower-dimensional mapping $\phi(x) = W^\top x$ to predict quantiles via $\widehat{h}(x) = \langle \widehat{\theta}, \phi(x) \rangle$.

3. Compare the coverage of the full-conformal method, standard split conformal with a static threshold, meaning confidence sets of the form $\widehat{C}(x) = \{y \in [k] \mid s(x, y) \leq \widehat{\tau}\}$, and split conformal with the threshold function $\widehat{h}(x)$ fit on the validation data. To perform the comparison, we draw subsamples from the test $Z_{\text{test}} = \{(x_i, y_i)\}_{i=1}^{n_{\text{test}}}$ by defining groups $G$ of the form

$$G_{j,>} = \{(x, y) \in Z_{\text{test}} \mid \langle w_j, x \rangle \geq \mathsf{Quant}_{.8}(\{\langle w_j, x_i \rangle\}_{i=1}^{n_{\text{test}}})\} \quad \text{and}$$
$$G_{j,<} = \{(x, y) \in Z_{\text{test}} \mid \langle w_j, x \rangle \leq \mathsf{Quant}_{.2}(\{\langle w_j, x_i \rangle\}_{i=1}^{n_{\text{test}}})\}$$

for each row $w_j^\top$ of the random dimension reduction matrix $W$ from step 2.

Figure 1 displays the results of this experiment. In the figure, we notice three main results: first, the static thresholded sets $\widehat{C}(x) = \{y \mid s(x, y) \leq \widehat{\tau}\}$ have substantially more variability in coverage on the random slices of the dataset. Second, the split conformal method and full conformal methods have similar coverage on each of the slices, with some slices exhibiting more variability of the full conformal methodology and some with the split conformal methodology, but all around the nominal (desired) 90% coverage level. Finally, the split conformal methods slightly undercover marginally, while the full conformal method slightly overcovers marginally.

## 5  Conclusion and discussion

The results in our experiments, though they are relatively small scale, appear to be consistent with other experiments on regression that we present in Appendix D, where we also investigate the appropriate choice of the level $\alpha$ to obtain a desired coverage level coverage. In brief: split conformal methods with confidence sets using adaptive thresholds of the form $\widehat{C}(x) = \{y \mid s(x, y) \leq \widehat{h}(x)\}$ can indeed provide stronger coverage than non-adaptive thresholds. Moreover, they are *much* faster to compute with than full conformal methods—in the experiment in Figure 1, the split conformal

method was roughly $8000\times$ faster than the full conformal method. Additionally, they enjoy strong sample-conditional stability as well as minimax optimality.

In spite of this, when the adaptive threshold $\widehat{h}(x)$ comes from a class of functions that is high-dimensional relative to the size of the data available for calibration, these methods can undercover, as they exhibit downward bias in their coverage. This bias is easy to correct for a static threshold $\widehat{C}(x) = \{y \mid s(x,y) \leq \widehat{\tau}\}$ by simply using a slightly larger quantile, however, it is unclear how to address it in adaptive scenarios. This makes obtaining a data-adaptive way to compute coverage bias an open and important research question: can we avoid the $\sqrt{n}$ error in each of the conditional coverage guarantees? As Gibbs et al. [12, Thm. 2] show, "full conformal" methods have errors scaling as $O(d/n)$ in these cases, while the methods in the current paper have a downward coverage bias that appears to scale with $\sqrt{d/n}$. This is, of course, worse, and delineating the extent to which this bias matters remains open.

Certainly, the minimax lower bounds that Areces et al. [1] demonstrate show that it is impossible to achieve anything improving on $\sqrt{d/n}$ error for two-sided guarantees, but if we *only* wish to demonstrate weighted coverage lower bounds $\mathbb{P}_w(Y_{n+1} \in \widehat{C}_n(X_{n+1}) \mid P_n) \geq 1 - \alpha - O(1)\frac{d \log(1/\delta)}{n}$, then it may be possible to achieve one-sided guarantees without such errror. One approach, which Cauchois et al. [8] adopt, is to split the validation sample to "re-conformalize" the confidence sets, so that the only concentration one needs is on a single threshold being learned. This would mean fitting a confidence set $C(x) = \{y \mid s(x,y) \leq \widehat{h}(x)\}$ on one validation sample, and on the second, modifying a single threshold $\widehat{\tau}$ to fit $\widehat{C}(x) = \{y \mid s(x,y) \leq \widehat{h}(x) + \widehat{\tau}\}$; this, however, would not provide any approximate conditional guarantees. It seems plausible that one might instead be able to use leave-one-out sampling to estimate this downward bias [11, 3]. Identifying such an offline correction without relying on asymptotic error calculations, as our heuristic development in Appendix D does, could make these procedures substantially more practical, by both enjoying the test-time speed of split conformal methods and the coverage accuracy of full-conformal procedures.

A second set of open questions relates to the experiments. While we have been careful to identify them as "exploratory" (because the main focus of this paper is theoretical), they do not directly address the sample-conditional aspects (i.e., coverage given $P_n$) of the guarantees so much as the "$\phi$–conditional" guarantees (or, perhaps, group-wise guarantees), as in Theorem 3 and Corollary 3.1. It would be (to the author at least) quite interesting to understand whether sample-conditional coverage is practically relevant, especially relative to the marginal coverage guarantees full-conformal inference provides. Bian and Barber [6] provide a counterexample showing that there exist cases where full-conformal methods are quite unstable. But understanding the extent to which this is practically meaningful would require experimental work that is beyond the current focus of this paper.

## Acknowledgments

This research was supported in part by the National Science Foundation under grant IIS-2006777 and the Office of Naval Research under grant ONR N00014-22-1-2669. I also wish to thank John Cherian for many stimulating conversations about this paper, the students in the Statistics 315a at Stanford in the winter of 2025, whose questions inspired this work, and several anonymous reviewers, whose investigation of the paper improved its presentation, helped to identify open questions, and (hopefully) led to a clearer paper.

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

## A    Sample conditional coverage revisited: proofs

As mentioned in Section 1.3, Proposition 1 provides a natural point of departure for developing more sophisticated coverage guarantees. We thus provide this elementary proof, then demonstrate the result using uniform convergence techniques. These uniform convergence guarantees—which form the basis for providing guarantees for approximate weighted coverage (Definition 1.1) also provide a two-sided bound on sample-conditional coverage:

**Corollary A.1.** *Assume the scores* $S_i = s(X_i, Y_i)$ *are distinct with probability 1. Then for any* $\gamma > 0$*, with probability at least* $1 - 2e^{-2n\gamma^2}$ *over the sample* $P_n$,

$$1 - \alpha - \gamma \leq \mathbb{P}(Y_{n+1} \in \widehat{C}_n(X_{n+1}) \mid P_n) \leq 1 - \alpha + \frac{1}{n} + \gamma.$$

### A.1    An elementary proof of Proposition 1

For the scalar random variable $S$, define the $\beta$-quantile

$$q^\star(\beta) := \inf \left\{ t \in \mathbb{R} \mid \mathbb{P}(S \leq t) \geq \beta \right\}. \tag{11}$$

Because the CDF is right continuous, we have $\mathbb{P}(S \leq q^\star(\beta)) \geq \beta$, and $\mathbb{P}(S > q^\star(\beta)) = 1 - \mathbb{P}(S \leq q^\star(\beta)) \leq 1 - \beta$. For $\gamma > 0$ and any $\tau \in \mathbb{R}$, the inequality

$$\mathbb{P}(S_{n+1} > \tau) > \alpha + \gamma, \text{ i.e. } \mathbb{P}(S_{n+1} \leq \tau) < 1 - \alpha - \gamma,$$

implies that $\tau < q^\star(1 - \alpha - \gamma)$.

Consider the event that $\widehat{\tau}_n < q^\star(1 - \alpha - \gamma)$. For this to occur, it must be the case that

$$\frac{1}{n} \sum_{i=1}^n 1\{S_i < q^\star(1 - \alpha - \gamma)\} \geq 1 - \alpha. \tag{12}$$

But this event is unlikely: define the Bernoulli indicator variables $B_i = 1\{S_i < q^\star(1 - \alpha - \gamma)\}$. Then $\mathbb{E}[B_i] \leq 1 - \alpha - \gamma$, and Hoeffding's inequality implies that $\overline{B}_n = \frac{1}{n} \sum_{i=1}^n B_i$ satisfies

$$\mathbb{P}\left(P_n(S < q^\star(1 - \alpha - \gamma)) \geq 1 - \alpha\right) = \mathbb{P}\left(\overline{B}_n \geq 1 - \alpha\right)$$
$$\leq \mathbb{P}\left(\overline{B}_n - \mathbb{E}[\overline{B}_n] \geq \gamma\right) \leq \exp(-2n\gamma^2).$$

That is,

$$\mathbb{P}\left(\widehat{\tau}_n < q^\star(1 - \alpha - \gamma)\right) \leq \exp\left(-2n\gamma^2\right)$$

for any $\gamma > 0$, and so we must have

$$\mathbb{P}\left(S_{n+1} > \widehat{\tau}_n \mid P_n\right) \leq \alpha + \gamma \text{ with probability at least } 1 - e^{-2n\gamma^2}.$$

Rearranging and recalling that $Y_{n+1} \notin \widehat{C}_n(X_{n+1})$ if and only if $s(X_{n+1}, Y_{n+1}) > \widehat{\tau}_n$, i.e., if $S_{n+1} > \widehat{\tau}_n$ gives the result.

### A.2    A proof of Proposition 1 using uniform convergence

Our alternative approach to the proof of Proposition 1 uses the bounded differences inequality and a uniform concentration guarantee. First, for any estimated threshold $\widehat{\tau}_n$, we have the trivial inequality

$$\mathbb{P}(S_{n+1} > \widehat{\tau}_n \mid P_n) = \mathbb{P}(S_{n+1} > \widehat{\tau}_n \mid P_n) - P_n(S > \widehat{\tau}_n) + P_n(S > \widehat{\tau}_n)$$
$$\leq \sup_{\tau \in \mathbb{R}} |P(S > \tau) - P_n(S > \tau)| + P_n(S > \widehat{\tau}_n).$$

Then because we choose $\widehat{\tau}_n$ so that $P_n(S \leq \widehat{\tau}_n) \geq 1 - \alpha$, we obtain

$$\mathbb{P}(S_{n+1} > \widehat{\tau}_n \mid P_n) \leq \alpha + \sup_{\tau \in \mathbb{R}} |P(S > \tau) - P_n(S > \tau)|. \tag{13a}$$

If the values $S_i$ are distinct, then $P_n(S \leq \widehat{\tau}_n) \leq 1 - \alpha + \frac{1}{n}$, and so a completely similar calculation yields

$$\mathbb{P}(S_{n+1} > \widehat{\tau}_n \mid P_n) \geq \alpha - \frac{1}{n} - \sup_{\tau \in \mathbb{R}} |P(S > \tau) - P_n(S > \tau)|. \tag{13b}$$

In either case, if we can control the deviation $|P(S > \tau) - P_n(S > \tau)|$ uniformly across $\tau$, we will have evidently proved the desired result.

We consider two arguments, the first yielding sharper constants, while the second generalizes to weighted coverage. For the first, we apply the Dvoretsky-Kiefer-Wolfowitz inequality [23]:

$$\mathbb{P}\left(\sup_{\tau \in \mathbb{R}} |P(S > \tau) - P_n(S > \tau)| \ge t\right) \le 2e^{-2nt^2}$$

for all $t \ge 0$. Combining the equations (13), we thus obtain that

$$\mathbb{P}(S_{n+1} > \widehat{\tau}_n \mid P_n) \le \alpha + \gamma \text{ with probability at least } 1 - 2e^{-2n\gamma^2}.$$

If the scores are distinct, the corresponding lower bound is immediate, giving Corollary A.1.

The final alternative argument to control the uniform deviations in the bounds (13) underpins our more sophisticated guarantees in the sequel, relying on uniform concentration guarantees and the Vapnik-Chervonenkis (VC) dimension. First, recall the classical bounded differences inequality [24, 32], where we say a function $f : \mathcal{X}^n \to \mathbb{R}$ satisfies $c_i$-bounded differences if

$$|f(x_1^{i-1}, x_i, x_i, x_{i+1}^n) - f(x_1^{i-1}, x_i', x_{i+1}^n)| \le c_i \text{ for all } x_1^{i-1}, x_{i+1}^n, x_i, x_i' \in \mathcal{X}.$$

**Lemma A.1** (Bounded differences). *Let $X_1, \ldots, X_n$ be independent random variables and $f$ satisfy $c_i$-bounded differences. Then for all $t \ge 0$,*

$$\max\left\{\mathbb{P}(f(X_1^n) - \mathbb{E}[f(X_1^n)] \ge t), \mathbb{P}(f(X_1^n) - \mathbb{E}[f(X_1^n)] \le -t)\right\} \le \exp\left(-\frac{2t^2}{\sum_{i=1}^n c_i^2}\right).$$

We then observe that $f(P_n) := \sup_{\tau \in \mathbb{R}} |P(S > \tau) - P_n(S > \tau)|$ trivially satisfies bounded differences. Indeed, let $P_n'$ differ from $P_n$ in a single observation. Then defining $\|P - P_n\|_\infty = \sup_\tau |P(S > \tau) - P_n(S > \tau)|$ for shorthand, we obtain

$$\left|\|P - P_n\|_\infty - \|P - P_n'\|_\infty\right| \le \|P_n - P_n'\|_\infty \le \frac{1}{n}$$

by the triangle inequality and that only one example may change. Lemma A.1 then implies

$$\mathbb{P}\left(\|P - P_n\|_\infty \ge \mathbb{E}[\|P - P_n\|_\infty] + t\right) \le e^{-2nt^2}$$

for $t \ge 0$, so that we need only control $\mathbb{E}[\|P - P_n\|_\infty]$. For this, we perform a standard symmetrization argument [e.g. 29, Ch. 2.3]: let $P_n^0 = \frac{1}{n} \sum_{i=1}^n \varepsilon_i \mathbf{1}_{S_i}$, where $\varepsilon_i \overset{\text{iid}}{\sim} \mathsf{Uni}\{\pm 1\}$ are i.i.d. Rademacher variables and $\mathbf{1}_{S_i}$ denotes a point mass on $S_i$. By introducing independent copies of $S_i$ and applying Jensen's inequality [29, Lemma 2.3.1], we have the bound

$$\mathbb{E}\left[\|P_n - P\|_\infty\right] \le 2\mathbb{E}\left[\|P_n^0\|_\infty\right] = 2\mathbb{E}\left[\sup_{\tau \in \mathbb{R}} \left|\frac{1}{n} \sum_{i=1}^n \varepsilon_i 1\{S_i > \tau\}\right|\right].$$

Because the class of functions $s \mapsto 1\{s > \tau\}$ has VC-dimension at most 1, Dudley's entropy integral (see, e.g. [29, Corollary 2.2.8 and Thm. 2.6.7] or [32, Eq. (5.5.1)]) shows that

$$\mathbb{E}\left[\|P_n^0\|_\infty\right] \le \frac{c}{\sqrt{n}}$$

for a numerical constant $c$. We then obtain that for any $\gamma \ge 0$,

$$\mathbb{P}\left(S_{n+1} > \widehat{\tau}_n \mid P_n\right) \le \alpha + \frac{c}{\sqrt{n}} + \gamma \text{ w.p. } 1 - e^{-2n\gamma^2}$$

by the inequalities (13), where $c$ is a numerical constant.

## A.3 Proof of Proposition 2

Recall the quantile mapping $q^\star$ from the definition (11) and that for fixed $\gamma \in [0, \alpha]$, the event $\widehat{\tau}_n < q^\star(1 - \alpha - \gamma)$ can occur only if $P_n(S < q^\star(1 - \alpha - \gamma)) \ge 1 - \alpha$. Then defining $B_i = 1\{S_i < q^\star(1 - \alpha - \gamma)\}$ and recalling inequality (12), we obtain

$$\mathbb{P}(\widehat{\tau}_n < q^\star(1 - \alpha - \gamma)) \le \mathbb{P}(\overline{B}_n \ge 1 - \alpha) = \mathbb{P}(\overline{B}_n - \mathbb{E}[\overline{B}_n] \ge 1 - \alpha - \mathbb{E}[\overline{B}_n]).$$

Define $p(\gamma) = \mathbb{E}[B_i] = \mathbb{P}(S < q^\star(1 - \alpha - \gamma)) < 1 - \alpha - \gamma$, so that $t = t(\gamma) := 1 - \alpha - p(\gamma) > \gamma$. Then $\mathrm{Var}(B_i) = p(\gamma)(1 - p(\gamma)) = (\alpha + t)(1 - \alpha - t)$, and Bernstein's inequality implies

$$\mathbb{P}(\overline{B}_n - \mathbb{E}[\overline{B}_n] \geq t) \leq \exp\left(-\frac{nt^2}{2(1 - \alpha - t)(\alpha + t) + \frac{2}{3}t}\right)$$

$$= \exp\left(-\frac{nt^2}{2(1 - \alpha)\alpha + (\frac{8}{3} - 4\alpha)t - t^2}\right) \leq \exp\left(-\frac{nt^2}{2(1 - \alpha)\alpha + \frac{8}{3}t}\right).$$

Notably, $t \mapsto \frac{nt^2}{2(1-\alpha)\alpha + \frac{8}{3}t}$ is increasing in $t$, so that

$$\mathbb{P}(\widehat{\tau}_n < q^\star(1 - \alpha - \gamma)) \leq \exp\left(-\frac{n\gamma^2}{2(1 - \alpha)\alpha + \frac{8}{3}\gamma}\right).$$

If the scores $S$ have a density, $\mathbb{P}(S \leq q^\star(\beta)) = \beta$ for any $\beta \in (0, 1)$. Then we may also consider the event that $\widehat{\tau}_n > q^\star(1 - \alpha + \gamma)$. For this to occur, we require

$$P_n(S < q^\star(1 - \alpha + \gamma)) \leq 1 - \alpha,$$

and defining $B_i = 1\{S_i < q^\star(1 - \alpha + \gamma)\}$, we have $\mathbb{E}[B_i] = 1 - \alpha + \gamma$ and so

$$\mathbb{P}(\overline{B}_n \leq 1 - \alpha) = \mathbb{P}(\overline{B}_n - \mathbb{E}[\overline{B}_n] \leq -\gamma) \leq \exp\left(-\frac{n\gamma^2}{2(1 - \alpha + \gamma)(\alpha - \gamma) + \frac{2}{3}\gamma}\right)$$

$$\leq \exp\left(-\frac{n\gamma^2}{2(1 - \alpha)\alpha + \frac{2}{3}\gamma}\right)$$

for $\gamma \in [0, \alpha]$. Combining the two cases, for $\gamma \geq 0$ we have

$$\max\left\{\mathbb{P}\left(\widehat{\tau}_n < q^\star(1 - \alpha - \gamma)\right), \mathbb{P}\left(\widehat{\tau}_n > q^\star(1 + \alpha + \gamma)\right)\right\} \leq \exp\left(-\frac{n\gamma^2}{2\alpha(1 - \alpha) + \frac{8}{3}\gamma}\right).$$

Solving to guarantee the right hand side is at most $\delta$ yields

$$\gamma_n := \frac{4\log\frac{1}{\delta}}{3n} + \sqrt{\left(\frac{4}{3n}\log\frac{1}{\delta}\right)^2 + \frac{2\alpha(1 - \alpha)}{n}\log\frac{1}{\delta}}.$$

Applying a union bound implies Proposition 2.

# B  Proof of Theorems 2 and 3

We split the proof into three sections, first giving the shared building blocks, then specializing.

## B.1  Proof of Theorems 2 and 3: building blocks

Our proof leverages a combination of Talagrand's concentration inequalities for empirical processes, a VC-dimension calculation, and localized Rademacher complexities [5, 19]. We begin with the form of Talagrand's empirical process inequality with constants due to Bousquet [7].

**Lemma B.1** (Talagrand's empirical process inequality). *Let $\mathcal{F}$ be a countable class of functions with $Pf = 0$ and $\|f\|_\infty \leq b$ for $f \in \mathcal{F}$. Let $Z = \sup_{f \in \mathcal{F}} P_n f$ and $\sigma^2 = \sigma^2(\mathcal{F}) = \sup_{f \in \mathcal{F}} Pf^2$. Define $v^2 := \sigma^2 + 2b\mathbb{E}[Z]$. Then for $t \geq 0$,*

$$\mathbb{P}\left(Z \geq \mathbb{E}[Z] + \sqrt{2v^2 t} + b\frac{t}{3}\right) \leq e^{-nt}.$$

Because we will consider functions of the form $f(x, s) = \langle u, \phi(x)\rangle 1\{\langle \theta, \phi(x)\rangle > s\}$, we also require some control over the complexity of such rank-one-like products.

**Lemma B.2.** *Let $\mathcal{H}$ and $\mathcal{G}$ be classes of functions with VC-dimensions $d_1$ and $d_2$, respectively. Then the classes of functions*

$$\mathcal{F} := \{f \mid f(x) = g(x)1\{h(x) > 0\}\} \quad \text{and} \quad \mathcal{F}_+ := \{f \mid f(x) = g(x)1\{h(x) > 0\} - cg(x)\}$$

*where $c$ is a constant have VC-dimension $O(1)(d_1 + d_2)$.*

**Proof** For a set of points $x_1, \ldots, x_n$, let $\mathcal{S}(x_1^n, \mathcal{H}) = \{1\{h(x_i) > 0\}\}_{i=1}^n$ be the set of "sign" vectors that $h$ realizes. By definition of the VC-dimension and the Sauer-Shelah lemma, this set has cardinality at most $\sum_{i=0}^{d_1} \binom{n}{i} \leq (\frac{ne}{d_1})^{d_1}$. Similarly, the set of signs

$$\mathcal{S}(x_1^n, \mathcal{F}) = \{\text{sign}(g(x_i)) \cdot 1\{h(x_i) > 0\}\}_{i=1}^n$$

has cardinality at most $\sum_{i=0}^{d_1} \binom{n}{i} \cdot \sum_{i=0}^{d_2} \binom{n}{i} \leq (\frac{ne}{d_1})^{d_1} (\frac{ne}{d_2})^{d_2}$. If $n$ is large enough that

$$\left(\frac{ne}{d_1}\right)^{d_1} \cdot \left(\frac{ne}{d_2}\right)^{d_2} < 2^n,$$

then certainly $\mathcal{F}$ cannot shatter $n$ points; this occurs once $n \geq c \cdot (d_1 + d_2)$ for some numerical constant $c$. For the second class the argument is similar. $\square$

For the next lemma, our main technical building block for convergence, we consider the class of functions $\mathcal{F}$ indexed by $u \in \mathbb{B}$ and $h \in \mathcal{H}$, where $\mathcal{H}$ is a class with VC-dimension at most $d$, with

$$f(x, s) = f_{u,h}(x, s) := \langle u, \phi(x) \rangle 1\{s > h(x)\}. \tag{14}$$

Each of these functions evidently satisfies $|f(x, s)| \leq b_\phi$. The variance proxy

$$v^2(u, h) := v^2(f_{u,h}) = P f_{u,h}(X, S)^2 = \mathbb{E}[\langle \phi(X), u \rangle^2 1\{S > h(X)\}] \tag{15}$$

and its empirical variant

$$v_n^2(u, h) = P_n f_{u,h}(X, S)^2 = \frac{1}{n} \sum_{i=1}^n \langle \phi(X_i), u \rangle^2 1\{S_i > h(X_i)\}.$$

will allow us to bound deviations of $P_n f$ from $P f$ relative to $v^2(f)$.

For later use, we recall the *empirical Rademacher complexity* of a function class $\mathcal{F}$,

$$\mathfrak{R}_n(\mathcal{F}) := \frac{1}{n} \mathbb{E}\left[\sup_{f \in \mathcal{F}} \sum_{i=1}^n \varepsilon_i f(X_i) \mid X_1^n\right],$$

where $\varepsilon_i \overset{\text{iid}}{\sim} \text{Uni}\{\pm 1\}$ are random signs. In some cases, we will require *localized Rademacher complexities* [5, 19] around a class $\mathcal{F}_r := \{f \mid P f^2 \leq r^2\}$, which contains functions of small variance, allowing us to "relativize" bounds. Bartlett et al. [5, Proof of Corollary 3.7] demonstrate the following.

**Lemma B.3.** *Let $\mathcal{F}$ be a star-convex collection of functions, meaning that $f \in \mathcal{F}$ implies $\lambda f \in \mathcal{F}$ for $\lambda \in [0, 1]$, and assume that $\sup_x |f(x)| \leq b$ and $\mathcal{F}$ has VC-dimension $d$. Then*

$$\mathbb{E}\left[\mathfrak{R}_n(\mathcal{F}_r)\right] \leq \frac{b_\phi}{n} + cr\sqrt{\frac{d}{n} \log \frac{b_\phi}{r}} \text{ if } r^2 > b_\phi^2 \frac{d}{n} \log \frac{n}{d}, \tag{16}$$

*where $c < \infty$ is a numerical constant.*

We will combine the VC-bound in Lemma B.2, the version of Talagrand's empirical process inequality in Lemma B.1, and a localization argument on Rademacher complexities via inequality (16) to prove the following lemma in Appendix C.2.

**Lemma B.4.** *Let $\mathcal{F}$ be the class of functions (14). Let $K_n = 1 + \log_2 n$. Then for all $t \geq 0$, with probability at least $1 - K_n e^{-t}$ over the draw of the sample $P_n$,*

$$|(P_n - P)f| \leq c\left[v(f)\sqrt{\frac{d \log n + t}{n}} + b_\phi \frac{d \log n + t}{n}\right]$$

*simultaneously for all $f \in \mathcal{F}$, where $c < \infty$ is a numerical constant. In addition, with the same probability,*

$$|(P_n - P)\langle u, \phi(X) \rangle| \leq c\left[\sqrt{P\langle u, \phi \rangle^2}\sqrt{\frac{d \log n + t}{n}} + b_\phi \frac{d \log n + t}{n}\right]$$

*simultaneously for all $u \in \mathbb{B}$.*

Next we present a version of a result appearing as [32, Theorem 14.12] (the result there assumes functions are mean-zero, but an inspection of the proof shows this is unnecessary); see also the results of [25] and [9, Proof of Proposition 1]. These show that second moments satisfy one-sided concentration bounds with high probability as soon as we have the fourth moment condition

$$\mathbb{E}[f^4(X, S)] \leq b^2 \mathbb{E}[f^2(X, S)] \text{ for all } f \in \mathcal{F}. \tag{17}$$

For the setting we consider, where $\mathcal{F}$ consists of product functions (14), inequality (17) immediately holds with $b = b_\phi$, though tighter constants may be possible.

**Lemma B.5.** *There exist numerical constants $0 < c$ and $C < \infty$ such that the following holds. Let inequality (17) hold and for $\mathcal{F}_r = \{f \mid Pf^2 \leq r^2\}$, let $r$ satisfy $\mathbb{E}[\mathfrak{R}_n(\mathcal{F}_r)] \leq \frac{r^2}{Cb}$. Then with probability at least $1 - e^{-cnr^2/b^2}$,*

$$P_n f^2 \geq \frac{1}{2} P f^2 \text{ simultaneously for all } f \text{ s.t. } v(f) \geq r.$$

Inequality (16) shows the conclusions of of Lemma B.5 hold if the radius $r$ satisfies

$$r\sqrt{\frac{d}{n} \log \frac{n}{d}} \lesssim \frac{r^2}{b_\phi} \text{ or } r^2 \gtrsim b_\phi^2 \cdot \frac{d}{n} \log \frac{n}{d}.$$

We then obtain the following consequence:

**Lemma B.6.** *Let $r^2 \gtrsim b_\phi^2 \frac{d}{n} \log \frac{n}{d}$. Then with probability at least $1 - e^{-cnr^2/b_\phi^2}$,*

$$P_n \langle u, \phi(X) \rangle^2 1\{S > h(X)\} \geq \frac{1}{2} P \langle u, \phi(X) \rangle^2 1\{S > h(X)\}$$

*simultaneously over $u \in \mathbb{B}$ and $h$ such that $P \langle u, \phi(X) \rangle^2 1\{S > h(X)\} \geq r^2$.*

Now, let $\widehat{h} = \langle \widehat{\theta}, \phi(\cdot) \rangle$, where $\widehat{\theta}$ solves the problem (8). Then simultaneously for all $u \in \mathbb{B}$, with probability at least $1 - K_n e^{-t}$,

$$\left| (P_n - P) \langle u, \phi(X) \rangle 1\left\{S > \widehat{h}(X)\right\} \right| \leq c \left[ v(\widehat{h}, u) \sqrt{\frac{d \log n + t}{n}} + b_\phi \frac{d \log n + t}{n} \right] \tag{18}$$

by Lemma B.4. Moreover, for $r^2 \gtrsim b_\phi^2 \frac{d}{n} \log \frac{n}{d}$, either $v(\widehat{h}, u) \leq r$ or

$$v^2(\widehat{h}, u) \leq 2 P_n \langle \phi(X), u \rangle^2 1\left\{S > \widehat{h}(X)\right\}$$

by Lemma B.6 (with the appropriate probability $1 - e^{-cr^2/b_\phi^2}$).

## B.2  Proof of Theorem 2: nonnegative weights

We now specialize our development to the particular cases that $\langle u, \phi(x) \rangle \geq 0$ for all $x \in \mathcal{X}$. First, we leverage the particular structure of the quantile loss to give a non-probabilistic bound on the empirical weights.

**Lemma B.7.** *Let $u$ be such that $\langle u, \phi(x) \rangle \geq 0$ for all $x \in \mathcal{X}$. Then*

$$P_n \langle \phi(X), u \rangle 1\left\{S > \widehat{h}(X)\right\} \leq \alpha P_n \langle \phi(X), u \rangle.$$

*If additionally $S_i$ are all distinct, then*

$$P_n \langle \phi(X), u \rangle 1\left\{S > \widehat{h}(X)\right\} \geq \alpha P_n \langle \phi(X), u \rangle - b_\phi(u) \frac{d}{n}.$$

**Proof**    The directional derivative $\ell'_\alpha(t; 1) := \lim_{\delta \downarrow 0} \frac{\ell_\alpha(t+\delta) - \ell_\alpha(t)}{\delta} = 1\{t \geq 0\} - (1 - \alpha)$. Then

$$P_n \langle \phi(X), u \rangle 1\left\{S > \widehat{h}(X)\right\} = P_n \langle \phi(X), u \rangle \left(1 - \alpha - 1\left\{S \leq \widehat{h}(X)\right\}\right) + P_n \langle u, \phi(X) \rangle \alpha$$

$$= P_n \langle \phi(X), u \rangle \left(-\ell'_\alpha(\widehat{h}(X) - S; 1)\right) + \alpha P_n \langle u, \phi(X) \rangle.$$

Letting $L_n(h) = P_n \ell_\alpha(h(X) - S)$, we now use that directional derivatives are positively homogeneous [15] and that by assumption $\widehat{h}$ minimizes $P_n \ell_\alpha(h(X) - S)$ over functions of the form $h(x) = \langle \theta, \phi(x) \rangle$ to obtain

$$P_n \langle \phi(X), u \rangle \left( -\ell'_\alpha(\widehat{h}(X) - S; 1) \right) = -P_n \ell'_\alpha(\widehat{h}(X) - S; \langle \phi(X), u \rangle) = -L'_n(\widehat{h}(X); u).$$

But of course, as $\widehat{h}$ minimizes $L_n$, we have $L'_n(\widehat{h}(X); u) \geq 0$ for all $u$, and so

$$P_n \langle \phi(X), u \rangle 1\left\{ S > \widehat{h}(X) \right\} \leq \alpha P_n \langle u, \phi(X) \rangle.$$

If $S_i$ are all distinct, then considering the left directional derivative, we also have

$$P_n \langle \phi(X), u \rangle 1\left\{ S \geq \widehat{h}(X) \right\} \geq \alpha P_n \langle u, \phi(X) \rangle.$$

If $\mathcal{I}_0 = \{i \mid \widehat{h}(X_i) = S_i\}$, then $\text{card}(\mathcal{I}_0) \leq d$, and so

$$0 \geq P_n \langle \phi(X), u \rangle \left( 1\left\{ S > \widehat{h}(X) \right\} - 1\left\{ S \geq \widehat{h}(X) \right\} \right) = -P_n \langle \phi(X), u \rangle 1\left\{ S = \widehat{h}(X) \right\} \geq -b_\phi(u) \frac{d}{n}.$$

Rearranging and performing a bit of algebra, we obtain the second claim of the lemma. $\qquad \square$

From the lemma, we see that

$$\begin{aligned}
& P \langle u, \phi(X) \rangle \left( 1\left\{ S > \widehat{h}(X) \right\} - \alpha \right) \\
&= (P - P_n) \langle u, \phi(X) \rangle \left( 1\left\{ S > \widehat{h}(X) \right\} - \alpha \right) + P_n \langle u, \phi(X) \rangle \left( 1\left\{ S > \widehat{h}(X) \right\} - \alpha \right) \\
&\leq (P - P_n) \langle u, \phi(X) \rangle 1\left\{ S > \widehat{h}(X) \right\} - \alpha(P - P_n) \langle u, \phi(X) \rangle \qquad (19)
\end{aligned}$$

by Lemma B.7. Additionally, the lemma implies that

$$P_n \langle \phi(X), u \rangle^2 1\left\{ S > \widehat{h}(X) \right\} \leq b_\phi(u) P_n \langle \phi(X), u \rangle 1\left\{ S > \widehat{h}(X) \right\} \leq b_\phi(u) \alpha P_n \langle \phi(X), u \rangle.$$

We use this to control the first term in the expansion (19) by combining these bounds with inequality (18) and considering that $v(\widehat{h}, u) \leq r$ or $v(\widehat{h}, u) > r$ where $r^2 = O(1) b_\phi^2 \frac{d}{n} \log \frac{n}{d}$. In the latter, we have $v^2(\widehat{h}, u) \leq c b_\phi(u) \alpha P_n \langle \phi(X), u \rangle$. We have therefore shown that for any $r^2 \gtrsim \frac{d}{n} \log \frac{n}{d}$, with probability at least $1 - K_n e^{-t} - e^{-nr^2}$, for all $u \in \mathbb{B}$ with $\langle u, \phi(x) \rangle \geq 0$,

$$\left| (P_n - P) \langle u, \phi(X) \rangle 1\left\{ S > \widehat{h}(X) \right\} \right| \qquad (20)$$
$$\leq c \left[ \left( \sqrt{b_\phi(u) \alpha P_n \langle u, \phi(X) \rangle} + b_\phi r \right) \sqrt{\frac{d \log n + t}{n}} + b_\phi \frac{d \log n + t}{n} \right].$$

Applying Lemma B.4 to the quantity $P_n \langle u, \phi(X) \rangle$ shows that simultaneously for all $u \in \mathbb{B}$,

$$|(P_n - P) \langle u, \phi(X) \rangle| \leq c \left[ \sqrt{b_\phi(u) P \langle u, \phi(X) \rangle} \sqrt{\frac{d \log n + t}{n}} + b_\phi \frac{d \log n + t}{n} \right]$$

with probability at least $1 - K_n e^{-t}$. Substituting this into the bounds (19) and (20), and ignoring lower-order terms (because $\alpha \leq 1$), we obtain the guarantee that for all $t \geq 0$ and $r^2 \gtrsim \frac{d}{n} \log \frac{n}{d}$, then with probability at least $1 - 2K_n e^{-t} - e^{-nr^2}$, for all $u$ such that $P \langle u, \phi(X) \rangle \geq b_\phi \frac{d \log n + t}{n}$,

$$\mathbb{E} \left[ \langle u, \phi(X) \rangle \left( 1\left\{ S > \widehat{h}(X) \right\} - \alpha \right) \mid P_n \right] \leq c \left[ \left( \sqrt{\alpha b_\phi(u) P \langle u, \phi(X) \rangle} + b_\phi r \right) \sqrt{\frac{d \log n + t}{n}} + b_\phi \frac{d \log n + t}{n} \right]$$

and for all $u$ such that $P \langle u, \phi(X) \rangle \leq b_\phi \frac{d \log n + t}{n}$,

$$\mathbb{E} \left[ \langle u, \phi(X) \rangle \left( 1\left\{ S > \widehat{h}(X) \right\} - \alpha \right) \mid P_n \right] \leq c b_\phi \left[ r \sqrt{\frac{d \log n + t}{n}} + \frac{d \log n + t}{n} \right].$$

Combining the inequalities and replacing $r^2$ with $\frac{d \log n + t}{n}$ gives the first claim of Theorem 2.

For the second claim, when the scores $S_i$ are distinct, note simply that we may replace inequality (19) with

$$
\begin{aligned}
&P\langle u, \phi(X)\rangle \left(1\left\{S > \widehat{h}(X)\right\} - \alpha\right) \\
&= (P - P_n)\langle u, \phi(X)\rangle \left(1\left\{S > \widehat{h}(X)\right\} - \alpha\right) + P_n\langle u, \phi(X)\rangle \left(1\left\{S > \widehat{h}(X)\right\} - \alpha\right) \\
&\geq (P - P_n)\langle u, \phi(X)\rangle 1\left\{S > \widehat{h}(X)\right\} - \alpha(P - P_n)\langle u, \phi(X)\rangle - b_\phi(u)\frac{d}{n}.
\end{aligned}
$$

The remainder of the argument is, *mutatis mutandis*, identical to the proof of the first claim of the theorem.

### B.3  Proof of Theorem 3: distinct scores

Because of the distinctness of $S_i$ and that we assume $\phi_1(x) = 1$ (that is, we include the constant offset), the optimality conditions for the quantile loss imply that

$$
\frac{d}{n} \geq \sum_{i=1}^{n} 1\left\{S_i > \widehat{h}(X_i)\right\} - \alpha \geq -\frac{d}{n}.
$$

So if $b_\phi(u) := \sup_{x \in \mathcal{X}} |\langle u, \phi(x)\rangle|$, then

$$
P_n \langle \phi(X), u\rangle^2 1\left\{S > \widehat{h}(X)\right\} \leq b_\phi^2(u)\left(\alpha + \frac{d}{n}\right).
$$

Applying inequality (18), we find that with probability at least $1 - K_n e^{-t} - e^{-cnr^2/b_\phi^2}$,

$$
\left|(P_n - P)\langle u, \phi(X)\rangle 1\left\{S > \widehat{h}(X)\right\}\right| \leq c\left[\left(b_\phi(u)\sqrt{\alpha} + r\right)\sqrt{\frac{d \log n + t}{n}} + b_\phi \frac{d \log n + t}{n}\right].
$$

The deviations $\alpha(P_n - P)\langle u, \phi(X)\rangle$ are of smaller order than this by Lemma B.4.

## C  Technical proofs

### C.1  Proof of Lemma 2.1

When $\mathbb{B}_2$ is the $\ell_2$-ball,

$$
\mathbb{E}\left[\sup_{h \in \mathcal{H}, u \in \mathbb{B}_2} \langle u, Z_n(h) - \mathbb{E}[Z_n(h)]\rangle\right] = \mathbb{E}\left[\sup_{h \in \mathcal{H}} \|Z_n(h) - \mathbb{E}[Z_n(h)]\|_2\right].
$$

Performing a typical symmetrization argument, we let $P_n^0 = \frac{1}{n}\sum_{i=1}^{n} \varepsilon_i \mathbf{1}_{X_i, S_i}$ be the symmetrized empirical measure, where $\varepsilon_i \overset{\text{iid}}{\sim} \mathsf{Uni}\{\pm 1\}$ are i.i.d. Rademacher variables, and define the symmetrized process $Z_n^0(h) = \frac{1}{n}\sum_{i=1}^{n} \varepsilon_i \phi(X_i) 1\{S_i > h(X_i)\}$. Then for the (random) set of vectors $\mathcal{V}_n = \{(1\{S_1 > h(X_1)\}, \ldots, 1\{S_n > h(X_n)\})\}_{h \in \mathcal{H}} \subset \{0,1\}^n$

$$
\mathbb{E}\left[\sup_{h \in \mathcal{H}} \|Z_n(h) - \mathbb{E}[Z_n(h)]\|_2\right] \leq 2\mathbb{E}\left[\sup_{h \in \mathcal{H}} \|Z_n^0(h)\|_2\right] \leq 2\mathbb{E}\left[\max_{v \in \mathcal{V}_n} \left\|\frac{1}{n}\sum_{i=1}^{n} \varepsilon_i \phi(X_i) v_i\right\|_2\right].
$$

Now, we recognize that because the vectors $\phi$ lie in a Hilbert space, we enjoy certain dimension free concentration guarantees. In particular, we have for any fixed $v \in \{0,1\}^n$ that

$$
\mathbb{P}\left(\left\|\sum_{i=1}^{n} \varepsilon_i \phi(X_i) v_i\right\|_2 \geq t \mid X_1^n\right) \leq 2\exp\left(-\frac{t^2}{2\Phi_n^2}\right),
$$

where $\Phi_n^2 := \sum_{i=1}^n \|\phi(X_i)\|_2^2$ by Pinelis [26, Theorem 3.5] (see also [16, Corollary 10]). In particular, using that for $U$ a nonnegative random variable $\mathbb{E}[U] = \int_0^\infty \mathbb{P}(U \geq u)du$, we obtain

$$\mathbb{E}\left[\max_{v \in \mathcal{V}_n} \left\|\sum_{i=1}^n \varepsilon_i \phi(X_i)v_i\right\|_2 \mid X_1^n\right] \leq \int_0^\infty \mathbb{P}\left(\max_{v \in \mathcal{V}_n} \left\|\sum_{i=1}^n \varepsilon_i \phi(X_i)v_i\right\|_2 \geq t \mid X_1^n\right) dt$$

$$\leq t_0 + 2\operatorname{card}(\mathcal{V}_n) \int_{t_0}^\infty \exp\left(-\frac{t^2}{2\Phi_n^2}\right) dt.$$

Recognizing the Gaussian tail bound that

$$\int_c^\infty e^{-\frac{t^2}{2\sigma^2}} dt = \sqrt{2\pi\sigma^2} \int_{c/\sigma}^\infty \frac{1}{\sqrt{2\pi}} e^{-z^2/2} dz \leq \sqrt{2\pi\sigma^2} \min\left\{\frac{1}{\sqrt{2\pi}}\frac{\sigma}{c}, 1\right\} \exp\left(-\frac{c^2}{2\sigma^2}\right)$$

by Mills' ratio, we see that for any $t_0 \geq 0$,

$$\mathbb{E}\left[\max_{v \in \mathcal{V}_n} \left\|\sum_{i=1}^n \varepsilon_i \phi(X_i)v_i\right\|_2 \mid X_1^n\right] \leq t_0 + 2\operatorname{card}(\mathcal{V}_n) \cdot \frac{\Phi_n^2}{t_0} \exp\left(-\frac{t_0^2}{2\Phi_n^2}\right).$$

Finally, recognize that $\mathcal{V}_n$ has cardinality at most $\left(\frac{en}{k}\right)^k$ by the Sauer-Shelah lemma because $\mathcal{H}$ has VC-dimension $k$. Consequently, we may take $t_0^2 = 2\log\operatorname{card}(\mathcal{V}_n)\Phi_n^2$ to obtain the bound

$$\mathbb{E}\left[\max_{v \in \mathcal{V}_n} \left\|\sum_{i=1}^n \varepsilon_i \phi(X_i)v_i\right\|_2 \mid X_1^n\right] \leq \sqrt{2\log\operatorname{card}(\mathcal{V}_n)}\Phi_n + \frac{\Phi_n}{\sqrt{2\log\operatorname{card}(\mathcal{V}_n)}} \leq 2\sqrt{k\log\frac{ne}{k}} \cdot \Phi_n.$$

Take expectations over $X_1^n$.

## C.2 Proof of Lemma B.4

For $r \geq 0$, define the localized class

$$\mathcal{F}_r := \left\{f \in \mathcal{F} \mid Pf^2 = \mathbb{E}[\langle v, \phi(X)\rangle^2 1\{S > h(X)\}] \leq r^2\right\}.$$

Note that $\mathcal{F}_r$ always includes the 0 function and is star-convex, because if $f \in \mathcal{F}_r$, then $\lambda f \in \mathcal{F}_r$ for $\lambda \in [0, 1]$. Recalling inequality (16), the second term dominates the first, and so

$$\mathbb{E}[\mathfrak{R}_n(\mathcal{F}_r)] \leq cr\sqrt{\frac{d}{n}\log\frac{n}{d}} \text{ if } r^2 \geq b_\phi^2 \frac{d}{n}\log\frac{n}{d}.$$

Define the random variable $Z_n(r) := \sup_{f \in \mathcal{F}_r}(P_n - P)f = \sup_{f \in \mathcal{F}_r}|(P_n - P)f|$, the equality following by symmetry of $\mathcal{F}_r$. Then Talagrand's concentration inequality (Lemma B.1) implies that

$$\mathbb{P}\left(Z_n(r) \geq \mathbb{E}[Z_n(r)] + \sqrt{2(r^2 + 2b_\phi\mathbb{E}[Z_n(r)])t} + \frac{b_\phi}{3}t\right) \leq e^{-nt}$$

for all $t \geq 0$. Applying a standard symmetrization argument and inequality (16), we thus obtain that for $r^2 \geq b_\phi^2 \frac{d}{n}\log\frac{n}{d}$, with probability at least $1 - e^{-t}$,

$$Z_n(r) \leq cr\sqrt{\frac{d\log n}{n}} + c\sqrt{\frac{r^2}{n^2} + \frac{b_\phi^2}{n}r\sqrt{\frac{d\log n}{n}}}\sqrt{t} + \frac{b_\phi t}{3n}.$$

As the last step, we apply a peeling argument [32, 28]: consider the intervals

$$E_k := \left(2^{k-1}\frac{b_\phi^2 d\log n}{n}, 2^k\frac{b_\phi^2 d\log n}{n}\right] \quad k = 1, 2, \ldots, K_n := \left\lceil \log_2\frac{n}{d\log n}\right\rceil.$$

Let $\mathcal{F}^k = \{f \in \mathcal{F} \mid Pf^2 \in E_k\}$, where $\mathcal{F}^0 = \{f \in \mathcal{F} \mid Pf^2 \leq \frac{d\log n}{n}\}$. Then evidently $\cup_{k=0}^{K_n}\mathcal{F}^k = \mathcal{F}$, and letting $r_k^2 = 2^k b_\phi^2 \frac{d\log n}{n}$, we have $\mathcal{F}_{r_k} \subset \mathcal{F}^k$. So by a union bound, with probability at least $1 - (K_n + 1)e^{-t}$,

$$Z_n(r_k) \leq cr_k\sqrt{\frac{d\log n}{n}} + c\sqrt{\frac{r_k^2}{n^2} + \frac{b_\phi}{n}\sqrt{\frac{r_k^2 d\log n}{n}}}\sqrt{t} + \frac{b_\phi}{3n}t \text{ for } k = 1, \ldots, K_n \qquad \text{(21a)}$$

and

$$Z_n(r_0) \le c\frac{d\log n}{n} + \sqrt{\frac{r_0^2}{n^2} + \frac{b_\phi}{n}\frac{d\log n}{n}}\sqrt{t} + \frac{b_\phi t}{3n}. \tag{21b}$$

Recall the definition $v^2(f) := Pf^2 = \mathrm{Var}(f) + (Pf)^2$. Then $f \in \mathcal{F}^k$ implies $\frac{1}{2}r_k \le v(f) \le r_k$, so that on the event that all the inequalities (21) hold, then simultaneously for all $f$ satisfying $v^2(f) \ge \frac{d\log n}{n}$, then

$$|(P_n - P)f| \le cv(f)\sqrt{\frac{d\log n}{n}} + c\sqrt{\frac{v^2(f)}{n^2} + \frac{b_\phi}{n}\sqrt{v^2(f)\frac{d\log n}{n}}}\sqrt{t} + \frac{b_\phi}{3n}t,$$

while for all $f$ with $v^2(f) \le \frac{d\log n}{n}$ we have

$$|(P_n - P)f| \le c\frac{d\log n}{n} + c\sqrt{\frac{d\log n}{n^3} + \frac{b_\phi}{n}\frac{d\log n}{n}}\sqrt{t} + \frac{b_\phi}{3n}t.$$

(To obtain the absolute bounds, we have used that $f \in \mathcal{F}$ implies $-f \in \mathcal{F}$ and each set $\mathcal{F}_r$ and $\mathcal{F}^k$ is symmetric.) Finally, we note that

$$\sqrt{\frac{v^2(f)}{n^2} + \frac{b_\phi}{n}\sqrt{v^2(f)\frac{d\log n}{n}}} \le \sqrt{\frac{v^2(f)}{n^2} + \frac{b_\phi^2 d\log n}{2n^2} + \frac{v^2(f)}{2n}}$$
$$\le \frac{b_\phi\sqrt{d\log n}}{\sqrt{2}n} + \frac{v(f)}{\sqrt{n}},$$

which implies the first statement of Lemma B.4.

The second statement follows via the same argument.

## D    Further simulations

We can incorporate a few recent theoretical results to enhance the practical performance of the proposed conformalization procedures, allowing some additional performance gains, while simultaneously exhibiting the need for future work. We consider mostly the difference between the full conformal approach that Gibbs et al. [12] develop and the split-conformal approaches that simply minimize the empirical loss (8). Our theoretical results provide no guidance to lower-order corrections to the desired level $\alpha$ to guarantee (exact) marginal coverage rather than approximate sample-conditional coverage, and so we proceed a bit heuristically here, using theoretical results to motivate modifications of the level $\alpha$ that do not change the sample-conditional coverage results we provide, but which turn out to be empirically effective.

To motivate our tweaks, recall the classical (unconditional) conformal approach to achieve exact finite-sample marginal coverage $\mathbb{P}(Y_{n+1} \in \widehat{C}_n(X_{n+1}))$, where the confidence confidence set $\widehat{C}_n(x) = \{y \mid s(x, y) \le \widehat{\tau}_n\}$. Setting $\widehat{\tau}_n = \mathsf{Quant}_{(1+1/n)(1-\alpha)}(S_1^n)$, the slightly enlarged quantile, guarantees $(1 - \alpha)$ coverage; this follows by letting $S_{(i,n)}$ be the order statistics of $S_1^n$ and $S_{(i,n+1)}$ those of $S_1^{n+1}$, and noting that the score $S_{n+1} \le S_{(k,n)}$ if and only if $S_{n+1} \le S_{(k,n+1)}$ [27, Lemma 2], so the inflation by $\frac{n+1}{n}$ is necessary. Equivalently, if we wish to achieve coverage $(1 - \alpha_{\mathrm{des}})$ using the estimator (8) with feature mapping $\phi(x) = 1$ fit at level $\alpha$, then $\alpha$ must solve $(1 - \alpha) = (1 + \frac{1}{n})(1 - \alpha_{\mathrm{des}})$, that is, $\alpha = 1 - (1 + \frac{1}{n})(1 - \alpha_{\mathrm{des}}) = (1 + \frac{1}{n})\alpha_{\mathrm{des}} - \frac{1}{n}$. That is, quantile regression under-covers.

When $\phi : \mathcal{X} \to \mathbb{R}^d$, it is then natural to heuristically imagine that the order statistics ought to "swap orders" by at most roughly $d$ items and so we ought to target coverage $\frac{n+d}{n}(1 - \alpha_{\mathrm{des}})$; unfortunately, it escapes our ability to prove such a result currently. Nonetheless, we consider a "naive" adaptation of the confidence level, setting $\alpha$ to solve

$$(1 - \alpha) = \left(1 + \frac{d}{n}\right)(1 - \alpha_{\mathrm{des}}), \text{ or } \alpha = \left(1 + \frac{d}{n}\right)\alpha_{\mathrm{des}} - \frac{d}{n}, \tag{22}$$

and then choosing $\widehat{\theta}$ to minimize (8) with this $\alpha$, which we term the "naive" choice. Bai et al. [2] give an alternative perspective, where they show that the actual marginal coverage achieved by quantile

regression at level $\alpha$ in the high-dimensional scaling $d, n \to \infty$ with $d/n \to \kappa \in (0, 1)$ is

$$(1 - \alpha) - \frac{d}{n}\left(\frac{1}{2} - \alpha\right) + o(d/n)$$

for $\alpha < \frac{1}{2}$, at least when the covariates are Gaussian. Solving this and ignoring the higher-order term, we recognize that to achieve desired coverage $\alpha$, we ought (according to this heuristic) to compute the estimator (8) using $\alpha$ solving

$$(1 - \alpha_{\mathrm{des}}) = (1 - \alpha) - \frac{d}{n}\left(\frac{1}{2} - \alpha\right) \quad \text{or} \quad \alpha = \frac{\alpha_{\mathrm{des}} - \frac{d}{2n}}{1 - \frac{d}{n}}. \tag{23}$$

We call the choice (23) the "scaling" choice. Neither of the rescalings (22) or (23) have any effect on the convergence guarantees our theory provides, as they are of lower order.

We perform two synthetic experiments that give a sense of the coverage properties of the methods we have analyzed. These exploratory experiments help provide justification for the heuristic corrections to the desired level $\alpha$ we set in the real data experiments.

### D.1 Level rescaling on a simple synthetic dataset

For our first experiment, we consider the simple setting of a standard Gaussian linear model, where we observe

$$y_i = \langle w^\star, x_i \rangle + \varepsilon_i, \quad \varepsilon_i \overset{\text{iid}}{\sim} \mathsf{N}(0, 1) \quad \text{and} \quad x_i \overset{\text{iid}}{\sim} \mathsf{N}(0, I_d).$$

We mimic the experiment Gibbs et al. [12, Fig. 3] provide, but we investigate the coverage properties of the coverage set from the estimator (8) with uncorrected $\alpha$ and level $\alpha$ corrected either naively (22) or via the scaling correction (23). In all cases, we use the feature map $\phi(x) = (1, 1\{x_1 > 0\}, \ldots, 1\{x_d > 0\}) \in \{0, 1\}^{d+1}$ indicating nonnegative coordinates. Figure 2 displays the results of this experiment for 1000 trials, where in each trial, we draw $w^\star \sim \mathsf{Uni}(\mathbb{S}^{d-1})$, fit a regression estimator $\widehat{w}$ on a training dataset of size $n_{\mathrm{train}} = 100$ using least squares, then conformalize this predictor using a validation set of size $n$ and evaluate its coverage on a test dataset of size $n_{\mathrm{test}} = 10n_{\mathrm{train}} = 1000$. We vary the ratio $n/d$ of the validation dataset, keeping $d = 20$ fixed. From the figure, it is clear that the uncorrected confidence set using $\alpha = \alpha_{\mathrm{des}} = .1$ undercovers, especially when the ratio $n/d < 20$ or so. The naive correction (22) appears to be a bit conservative, while the scaling correction (23) is more effective.

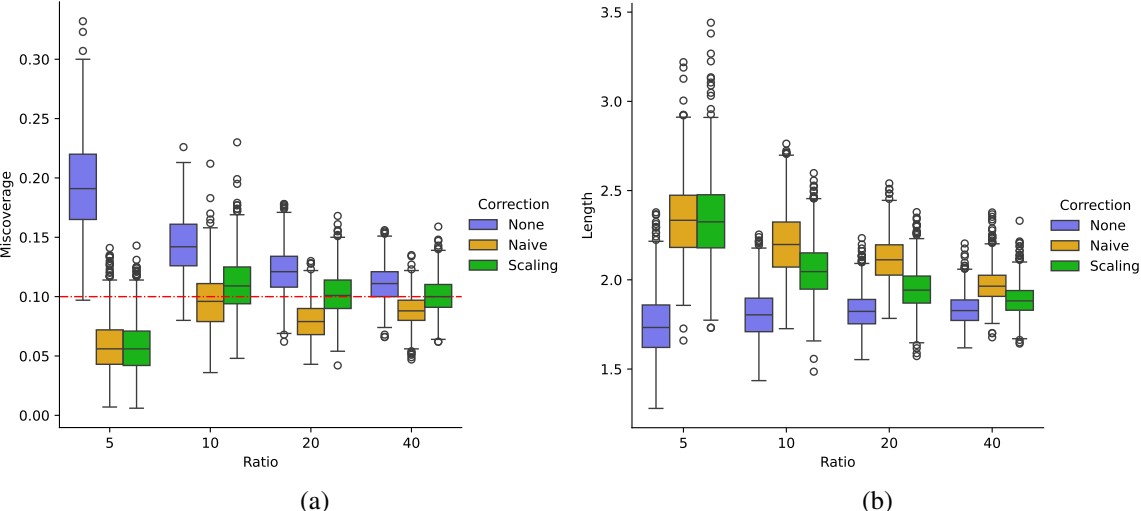

(a)                                                       (b)

**Figure 2.** Impact of the correction to $\alpha$ used in fitting the conformal predictor (8) for a desired level $\alpha_{\mathrm{des}} = .1$, i.e., 90% coverage. The "None" correction uses $\alpha = \alpha_{\mathrm{des}}$, "Naive" uses the correction (22), and "Scaling" uses the correction (23). (a) Coverage rates with the desired coverage marked as the red line. (b) Width of predictive intervals $\widehat{C}(x) = \{y \in \mathbb{R} \mid |\widehat{f}(x) - y| \le \widehat{\theta}^\top \phi(x)\}$.

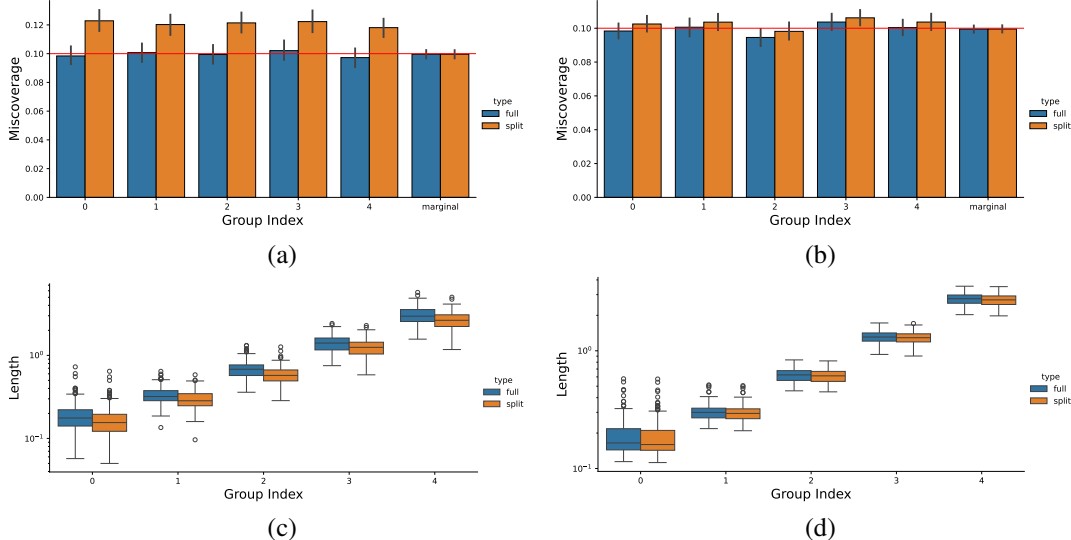

**Figure 3.** Comparison of full- and split-conformal methods on the simulated sinusoidal data of Sec. D.2 with $n_{\text{train}} = 200$ training examples and target miscoverage $\alpha = .1$. Plots (a) and (c) use validation sample sizes $n_{\text{val}} = 20k = 100$, while (b) and (d) use $n_{\text{val}} = 160k = 800$. Plots (a) and (b) show miscoverage $\mathbb{P}(Y \notin \widehat{C}(X) \mid X \in B_i)$ by group $B_i$; plots (c) and (d) prediction interval lengths.

## D.2 Full conformal versus split-conformal predictions

We briefly look at the coverage properties of the full conformalization method (5) from the paper [12], comparing with split-conformal methods (8), on a synthetic regression dataset we design to have asymmetric mean-zero heteroskedastic noise. We generate pairs $(X_i, Y_i) \in \mathbb{R}^2$ according to $Y = f(x) + \varepsilon(x)$, discretizing $x \in [0, 1]$ into bins $B_i = \{x \mid \frac{i}{k} \leq x < \frac{i+1}{k}\}$, $i = 0, \ldots, k - 1$, for $k = 5$. Within each experiment, we draw $U_0, U_1 \overset{\text{iid}}{\sim} \mathsf{Uni}[-1, 1]$ and $\phi_0, \phi_1 \overset{\text{iid}}{\sim} \mathsf{Uni}[\frac{\pi}{4}, 4\pi]$ to define

$$f(x) = U_0 \cos(\phi_0 \cdot x) + U_1 \sin(\phi_1 \cdot x).$$

Within the $i$th region $\frac{i}{k} \leq x < \frac{i+1}{k}$ we set $\lambda_{0,i} = \exp(3 - \frac{3}{k} i)$ and $\lambda_{1,i} = \exp(4 - \frac{3}{k} i)$, i.e., evenly spaced in $\{e^3, \ldots, e^0\}$ and $\{e^4, \ldots, e^1\}$, and draw

$$\varepsilon(x) \sim \begin{cases} \mathsf{Exp}(\lambda_{0,i}) & \text{with probability } \frac{\lambda_{0,i}}{\lambda_{0,i} + \lambda_{1,i}} = \frac{1}{1+e} \\ -\mathsf{Exp}(\lambda_{1,i}) & \text{otherwise,} \end{cases}$$

so that $\mathbb{E}[\varepsilon(x) \mid x] = 0$ but the noise is skewed upward, with variance increasing in $i$.

Figure 3 shows the results of this experiment over 200 independent trials, where in each experiment we draw a new mean function $f$ and fit it using a degree 5 polynomial regression on a training set of size $n_{\text{train}} = 200$. The conformalization methods use a group-indicator featurization $\phi(x) = (1, \mathbb{1}\{x \in B_1\}, \ldots, \mathbb{1}\{x \in B_k\})$ and confidence sets $C(x) = \{y \mid \theta_0^\top \phi(x) \leq y \leq \theta_1^\top \phi(x)\}$. Within each trial, we compute miscoverage proportions $\mathbb{P}(Y \notin \widehat{C}(X) \mid X \in B_i)$ for each bin $i$ on a test set of size 500, drawing a new function $f$. We vary the size of the validation data $n_{\text{val}} = \{10k, 20k, 40k, 80k, 160k\}$, and use the scaling correction (23) to set $\alpha$ for the split-conformal method. The figure plots results for validation sizes $40k$ and $160k$; from the figure—which is consistent with our other sample sizes and experiments—we see that when the validation size is large relative to the dimension of the mapping $\phi$, both methods are similar; for smaller ratios, the offline method undercovers slightly within the groups, though its marginal coverage remains near perfect in spite of the very non-Gaussian data.

We remark in passing that the full conformal method requires roughly $10\times$ the amount of time to compute predictions as the split conformal method requires to both fit a quantile prediction model and make its predictions. Once the split-conformal quantile model is available—it has been fit—this difference becomes roughly a factor of 2000–4000 in our experiments. For some applications, this

may be immaterial; for others, it may be a substantial expense, suggesting that a decision between the offline method and the online procedure may boil down to one of computational feasibility.

