# OpenReview forum: "Sample-Conditional Coverage in Split-Conformal Prediction"
_NeurIPS.cc/2025/Conference — NeurIPS 2025 poster_

### Official Review · Reviewer_538x · 2025-06-23

**Clarity:** 3
**Significance:** 2
**Originality:** 2
**Rating:** 4
**Confidence:** 3

**Summary:**

This paper presents new theoretical results demonstrating that split conformal methods can achieve near-nominal coverage with high probability.

**Questions:**

1. In line 245, please clarify the statement: "the static thresholded sets have more variability in coverage on the random slices of the dataset." A more intuitive explanation of the experiments results, especially in relation to the theoretical results, would be very helpful for understanding how the theory connects to the experimental observations.

2. Aside from group-conditional coverage, are there other types of conditional coverage that split conformal prediction can guarantee?

**Ethical Concerns:**

["NO or VERY MINOR ethics concerns only"]

**Final Justification:**

Thank you for these clarifications. And I keep my positive score of 4.

**Limitations:**

Yes, limitations are discussed in the last section of paper.

**Paper Formatting Concerns:**

No.

**Quality:**

3

**Strengths And Weaknesses:**

Strengths:
- The paper offers a fresh perspective on sample-conditional coverage for split conformal prediction methods, supported by strong theoretical guarantees and clear exposition.

Weaknesses:
- The experimental results are not sufficiently intuitive or easy to interpret.

---

> ### Author Rebuttal · Authors · 2025-07-30
>
> I thank the reviewer for the review and feedback.
>
> As the reviewer notes, the main purpose of this paper is theoretical, in that I attempt to provide the sharpest possible results on coverage and predictive inference with conformal-like methods. (Indeed, the results are minimax-optimal, so there is little room for improvement without further assumptions!)
>
> With regards to the experimental results, these are fair points. (See also my responses to the other reviewers, especially C1kY). Essentially, the experiment is showing that confidence sets of the form
> $$
> \hat{C}(x) = \\{y \mid s_y(x) \le \hat{\tau} \\}
> $$
> do not achieve coverage as close to the desired level $1 - \alpha$ (where $\alpha = .1$, i.e., 90% coverage) as do the adaptive sets
> $$
> \hat{C}(x) = \\{y \mid s_y(x) \le \hat{\theta}^T \phi(x)\\}.
> $$
> I will clarify this in revision. The latter sets--see Fig. 1--and the related sets that full-conformal inference gives appear to have coverage much closer to the desired 90% level.
>
> The reviewer also asks whether one can achieve things beyond group-conditional coverage. At some level, all conditional coverage is group-conditional (just with very small groups), but I think this is too pat an answer. By looking at linear combinations $\phi(x)^T \theta$ of group indicators, one can obtain coverage over unions of groups--still group-conditional, of course--or even more exotic setwise operations. Understanding the tradeoffs between pure distribution-free coverage and the types of conditional-like coverage one can achieve remains a tantalizing but active area of research (cf. the cited papers [10, 3, 28]).

---

> > ### Comment · Reviewer_538x · 2025-08-04
> > **Acknowledgement and response to the rebuttal**
> >
> > Thank you for these clarifications. And I keep my positive score of 4.

---

### Official Review · Reviewer_tt8B · 2025-06-23

**Clarity:** 2
**Significance:** 3
**Originality:** 3
**Rating:** 3
**Confidence:** 3

**Summary:**

This paper revisits the problem of constructing predictive confidence sets, aiming to achieve near desired coverage levels with high probability, a guarantee conditional on the validation data rather than marginal over it. The authors show that the natural method of performing quantile regression on a held-out dataset yields minimax optimal guarantees of coverage. In addition, the authors specifically consider coverage conditional on a covariate $X$ belonging to some group of interest. Numerical experiments are provided that are consistent with the positive theoretical results.

**Questions:**

1. I would appreciate it if the authors could further clarify the motivations and key contributions of this work, as this would help readers better appreciate these rigorous theoretical results in this paper.
2. I have some reservations regarding the experimental results. Could the authors clarify whether the demonstrated performance reflects sample-conditional coverage properties? From my perspective, these results appear to demonstrate group-conditional coverage properties rather than the claimed sample-conditional coverage.

Other minor comments:
● Typo: double appears in Line 136
● Typo: $\mathbb{S}^{d-1}$might be $\mathbb{B}^{d-1}$?

**Ethical Concerns:**

["NO or VERY MINOR ethics concerns only"]

**Final Justification:**

I have read through the responses and will maintain my score since the connection to sample-conditional coverage guarantees appears insufficiently developed in the experimental section.

**Limitations:**

Yes

**Paper Formatting Concerns:**

None.

**Quality:**

2

**Strengths And Weaknesses:**

**Strength**:
This work investigates the theoretical analysis of conditional coverage properties within the framework of split conformal prediction, and the authors demonstrate rigorous results and establish tighter bounds with solid theoretical proofs.

**Weaknesses**:
1. The authors presented lots of theoretical results as well as prior findings in related works, though the presentation might be a little confusing and could be more focused and clearer to highlight the key contributions better.
2. The authors focus on the sample-conditional coverage. A more thorough discussion of the underlying motivations could potentially enhance the applicability and practical value of this work.
3. This work appears to focus primarily on theoretical contributions (tighter bounds), while it doesn't include a new algorithm or approach.
4. The authors might consider providing more comprehensive elaboration on certain essential technical concepts. For example, the key results concerning minimax rate optimality may require additional explication for readers less familiar with this specialized framework.

---

> ### Author Rebuttal · Authors · 2025-07-30
>
> Thanks to the reviewer for the comments and feedback. Let me address the main points below (I will of course fix typos the reviewer notes):
>
> 1. Presentation. I'll do my best to tighten up the results; see also the response to reviewer D6i1. In brief, adding citations to existing corollaries should help make things clearer. If the reviewer thinks this would help, moving the proof of Theorem 1 later in the paper is certainly an option.
>
> 2. On sample conditional coverage, both experimental and philosophical (i.e., "Why do this at all?") questions: totally fair. As to the philosphical, I find Bian and Barber's [BB] paper "Training-conditional coverage for distribution-free predictive inference" instructive. Basically, in most ML applications, we wish to know that *given the training data we have,* we are likely to generalize well. Full-conformal methods cannot guarantee this, as [BB] shows; they only guarantee generalization in an average sense over both training and test data. As [BB] writes, "In practice, we are often interested in the coverage rate for test points once we fit a regression algorithm to a particular training set." I ought to have been clearer on that.
>
> On the experimental side, hoo-boy. The reviewer is absolutely right to point that out. The experiments--which, as I note, one ought to take more as "exploratory" (because the main focus of the paper is theoretical)--do not directly address the sample-conditional aspects of the guarantees so much as the "$\phi(x)$-conditional" guarantees (or, perhaps, group-wise guarantees). At this point, I will perhaps appeal to the mercy of the reviewers: it would be supremely interesting to understand whether sample-conditional coverage is practically relevant, especially relative to the marginal coverage guarantees full-conformal inference provides. [BB] provide a counterexample showing that there exist cases where full-conformal methods are quite unstable. But understanding the extent to which this is practically meaningful would require a slew of experimental work that is beyond the current focus of this submission. I'm of the (perhaps biased) opinion that a theoretically-focused paper should be allowed to be theoretical so as to keep the paper focused rather than spread too thin.
>
> 3. On the key technical underpinnings: I will (in revision) include more on the minimax optimality material. This is a good idea, and will hopefully help highlight more motivation for this paper. The paper by Areces et al. [1] is not so well-known. Basically, it states that if one wants to obtain $\mathcal{W}$-weighted conditional coverage, one must pay a factor involving both the desired confidence level $\alpha$ and VC-dimension $d$ of the class $\mathcal{W}$. Rewriting this material to highlight it as a lower bound--and connecting, e.g., to earlier work that does not achieve these optimal rates--will help the presentation, and I'll do that in revision.

---

> > ### Comment · Reviewer_tt8B · 2025-08-06
> >
> > Thank you for your detailed response and clarifications.
> >
> > Regarding the experimental results, while the current study demonstrates group-conditional coverage guarantees that connect with earlier theoretical findings in the paper, there appears to be insufficient linkage to the sample-conditional coverage guarantees (as also observed by Reviewer C1kY), given the paper's title. I would appreciate further discussion on this aspect.

---

### Official Review · Reviewer_C1kY · 2025-07-06

**Clarity:** 3
**Significance:** 2
**Originality:** 3
**Rating:** 3
**Confidence:** 3

**Summary:**

The paper studies Conformal Prediction under sample-conditional coverage, where the coverage holds with high probability conditional on the training and calibration data. The authors show that split conformal prediciton, including adaptive thresholds via quantile regression, can achieve approximate sample-conditional coverage. Theoretically results are established under standard i.i.d. assumptions, with minimax-optimal bounds in certain settings. The paper also includes a small empirical study to illustrate conditional coverage performance.

**Questions:**

- The theoretical guarantees for the threshold function depend on the VC dimension of the function class. In practice, if the model class is complex and the calibration sample is small, do the guarantees still hold meaningfully?

- The theory assumes i.i.d. data. In practical applications (e.g., covariate shift or posterior drift), how robust is the proposed method to deviations from this assumptions? Could the sample-conditional perspective extend to covariate shift scenarios?

- What does the empirical section actually validate? I don't see how this is related to the theoretical analysis.

**Ethical Concerns:**

["NO or VERY MINOR ethics concerns only"]

**Final Justification:**

I have read through the rebuttal and will maintain my score.

**Limitations:**

No. Please refer to the weaknesses and questions above.

**Paper Formatting Concerns:**

No.

**Quality:**

3

**Strengths And Weaknesses:**

Strengths:
- The theoretical results are solid and well-justified. Although i did not read all the proofs, they look good to me and the assumptions are reasonable.
- Conditional coverage is a meaningful and realistic goal to study, and the motivation behind focusing on it in this work is well explained.
- The presentation is mostly clear, and the theoretical contribution can be followed for readers familiar with the area.

Weaknesses:
- I don't quite get the idea of the empirical section. It is not connected to the theory. The setup is limited and does not clearly demonstrate the practical advantages of the proposed guarantees.
- While the viewpoint of this paper is well motivated, the method is based on existing works and conformal tools, and the novelty lies mainly in pure analysis rather than methodology.
- The paper may be hard to appreciate for readers unfamiliar with the conformal literature.

Typos:
- Line 136: duplicate "appears".
- Line 250: typo "slighlty"
- Line 261: typo "downard"
- Line 398: typo "shsared"
- The format of the reference part is inconsistent.

---

> ### Author Rebuttal · Authors · 2025-07-30
>
> Thanks to the reviewer for the thoughtful and careful review. A few responses here, focusing on the weaknesses the reviewer notes:
>
> The main feedback of the reviewer was to ask about the experiments. I could have been clearer in these. As the reviewer notes, the contributions of this paper are mostly on the theoretical side--giving optimal guarantees for predictive validity and sharp analyses--rather than empirical. Nonetheless, the point of the experiments was twofold:
> 1. To demonstrate that the offline conformal method (using X-dependent thresholds via $\hat{C}(x) = \\{y \mid s_y(x) \le \hat{\theta}^T \phi(x)\\}$) could provide closer to the target coverage of 90% than a single thresholded confidence set of the form $\hat{C}(x) = \\{y \mid s_y(x) \le \hat{\tau}\\}$ would. (So in Figure 1 of the submission, note that across random slices of the data, the split conformal method achieves much closer to 90% coverage than the static, i.e., single threshold confidence set.)
>
> 2. To see that an offline method could be computationally much more scalable than a full conformal method. The offline methods are about 8000x faster (at prediction time) than full conformal methods (see the comment in the discussion on page 9).
>
> The theoretical guarantees for the threshold function depend on the VC dimension of the function class. In practice, if the model class is complex and the calibration sample is small, do the guarantees still hold meaningfully?
>
> Bluntly, no. But to be fair, we wouldn't expect that to be the case: these methods provide distribution-free coverage, and so we cannot hope to do too much better (as the minimax lower bounds in reference [1] of the paper show). With that said, it would be interesting--I can include this as an open question--to give data-dependent guarantees. Another option, probably empirically as effective but which does not provide the same approximately conditional guarantees, is to split the validation sample to "re-conformalize" the confidence sets. Then the only concentration one needs is on the single threshold being learned. This would mean fitting a confidence set $C(x) = \\{y \mid s_x(y) \le \hat{h}(x)\\}$ on one validation sample, then on the second validation split, fitting $\hat{C}(x) = \\{y \mid s_x(y) \le \hat{h(x)} + \hat{\tau}\\}$, where only $\tau$ is fit on this second sample to correct for mistakes in $\hat{h}(x)$. (This is similar to the approach, e.g., in the paper "Knowing what You Know: valid and validated confidence sets in multiclass and multilabel prediction" by Cauchois et al., JMLR 2021.)
>
> The theory assumes i.i.d. data. In practical applications (e.g., covariate shift or posterior drift), how robust is the proposed method to deviations from this assumptions? Could the sample-conditional perspective extend to covariate shift scenarios?
>
> Good question. It depends on the structure of the covariate shift and the fidelity of the functionals being learned to the covariate shift. So, for example, if one fits a confidence set of the form $\hat{C}(x) = \{y \mid s(y, x) \le \hat{\theta}^T \phi(x)\}$, and the conditional distribution of $y$ is specified reasonably well by a linear functional of $\phi(x)$, then yes--the learned $\hat{C}(x)$ should transfer. If not, then we would expect failure. To be clear, this is the same strength/weakness of typical work on covariate shift: it really only makes sense insofar as there is no hidden confounding, that is, $x$ specifies the conditional distribution of $y$ completely. If it doesn't, for example, because of some unobserved variable $U$ with graphical structure $U \rightarrow Y \leftarrow X$, then frankly, covariate shift makes no sense anyway.
>
> Also, thanks for pointing out the typos; will address them. Though I confess I am not sure what the reviewer refers to when they write that "The format of the reference part is inconsistent." ?

---

> > ### Comment · Reviewer_C1kY · 2025-08-06
> >
> > Thank you for the detailed response. I have read through it and will maintain my score for now. Please disregard the earlier comment regarding reference.

---

### Official Review · Reviewer_D6i1 · 2025-07-22

**Clarity:** 3
**Significance:** 3
**Originality:** 3
**Rating:** 5
**Confidence:** 3

**Summary:**

This paper analyzes split conformal methods with confidence sets using adaptive thresholds of the form $\hat{C}(x) = \\{y | s(x, y) \leq \hat{h}(x)\\}$, where $\hat{h}$ is learnt via quantile regression on the calibration/validation data. It derives high-probability bounds for achieving approximate covariate-conditional coverage, while being conditioned on the calibration/validation data. Additionally, such methods are minimax rate optimal.

**Questions:**

__Questions and Comments__

1.	Did the authors run evaluations on regression tasks as well?

2.	It is not easy to demarcate existing theoretical results from new ones. Doing so will help with the presentation of the paper.

3.	What is $S_{1}^{n}$ in line 52?

__Typos__

1.	The equation in the panel under line 106 has two equalities with 0.

2.	"The construction (5) appears appears to obtain better..." $\rightarrow$ "The construction (5) appears to obtain better..." (line 136)

**Ethical Concerns:**

["NO or VERY MINOR ethics concerns only"]

**Final Justification:**

As suggested, the authors have run additional experiments on verifying the theoretical claims.

I remain in support of the paper!

**Limitations:**

Yes

**Quality:**

3

**Strengths And Weaknesses:**

__Strengths__

1.	Studying the conditional coverage of split conformal methods is an important research area. In practice, the calibration/validation is fixed, and covariate-conditional guarantees are ideal.

2.	The theoretical claims provide high-probability bounds for achieving approximate covariate-conditional coverage, while being conditioned on the calibration/validation data

3.	The paper is well written and has a good flow to it.

__Weaknesses__

1.	While the experimental results provide some evidence supporting the theoretical claims, is there a way to verify their correctness more concretely? For instance, by checking if the empirical coverage lies within the derived bounds?

---

> ### Author Rebuttal · Authors · 2025-07-29
>
> Thank you for the constructive review. Let me attempt to answer the reviewer's questions below:
>
> 1. While the experimental results provide some evidence supporting the theoretical claims, is there a way to verify their correctness more concretely? For instance, by checking if the empirical coverage lies within the derived bounds?
>
> Assuming the theorems are true (we all make mistakes, but I'm pretty confident on these), the empirical coverage will like within the derived bounds. While I cannot submit figures, I have done as the reviewer suggests--the bounds are conservative.
>
> Perhaps the more important question--which remains an open research question--is whether the \sqrt{n} error can be avoided. "Full conformal" methods have errors scaling as O(d/n) in these cases, while the methods in the current submission have a downward coverage bias that appears to scale with \sqrt{d/n}. This is, of course, worse. I ought to have noted this more carefully as an open question in the discussion. (It seems plausible that one might be able to *use* leave-one-out sampling to estimate the bias, but this needs more work.) Of course, the minimax lower bounds that Areces et al. (ref [1] in the submission) provide show that this is impossible for two-sided guarantees; if we only cared about coverage then it may be possible to achieve one-sided guarantees without the \sqrt{d/n} penalties.
>
> Other questions:
> 1. Did the authors run evaluations on regression tasks as well?
> Yes, though these appear only in appendices. The big-picture take-homes remain identical.
>
> 2. It is not easy to demarcate existing theoretical results from new ones. Doing so will help with the presentation of the paper.
> Apologies. In revision I'll make sure to incorporate known results *in* the result statements. To clarify briefly: Corollary 1.1, Proposition 1, and Corollary 2.1 are known; each other result in the paper is new. These new results include Proposition 2, Theorems 1--3, and their attendant lemmas and corollaries.
>
> 2. What is $S_{1}^{n}$ in line 52?
> It's supposed to stand for $S_1^n = (S_1, \ldots, S_n)$ where $S_i = s(X_i, Y_i)$, but I must've missed defining it... sorry!
>
> I'll fix all the noted typos.

---

> > ### Comment · Reviewer_D6i1 · 2025-08-08
> >
> > Thanks for your responses!
> >
> > The addition of the experiments on verifying the theoretical claims will further help the paper. I appreciate the authors including them.
> >
> > Also, please do include a discussion along the lines of your response above, as it could be an important research direction.
> >
> > I remain in support of the paper!

---

> > > ### Author Response · Authors · 2025-08-08
> > >
> > > Thanks much!

---

### Decision · Program_Chairs · 2025-09-17

**Decision:**

Accept (poster)

**Comment:**

The goal of this paper is to analyze split conformal prediction methods where the prediction sets/intervals are constructed using input-conditioned conformity score threshold, \hat{\tau}(x)), which learned using quantile regression on a calibration/validation set. The key contribution is to derive high-probability bounds for achieving approximate covariate-conditional coverage conditioned on the calibration/validation data.  Also, such methods have minimax optimal guarantees of coverage. The paper also provides sample experiments on a classification task in the main paper and regression tasks in the appendix to verify theory.

The paper had mixed ratings (two positive and two negative). The main concerns of the two negative reviewers' are:
1. Lack of methodological contribution
2. Lack of more experiments to demonstrate practical significance

The first one is minor as the focus of the paper is on theory. I acknowledge that the paper could be improved on the experiments side, but I'm convinced by the authors' response (which I found very refreshing and quite scientific in the current publication enterprise) to multiple reviewers' including D6i1 where they pointed to the fact that results on regression tasks are in the Appendix.

By considering all the factors, I'm leaning towards accepting the paper -- It is a solid theory paper on an important problem. I strongly encourage the authors' to make the following changes in the paper.
- Improve exposition of the paper to clearly articulate the motivation and take home message for practitioners.
- Clearly separate the known results and new results as suggested by one of the reviewer.
- Add more experiments as per the reviews and discussion using the extra page. Clearly articulate the goal and main findings of the experiments as done in your response.